# An injectable hydrogel enhances tissue repair after spinal cord injury by promoting extracellular matrix remodeling

Le Thi Anh Hong[1,2], Young-Min Kim [3], Hee Hwan Park [1,2], Dong Hoon Hwang[1], Yuexian Cui[1,2], Eun Mi Lee[1], Stephanie Yahn[4], Jae K. Lee[4], Soo-Chang Song [3,5] & Byung Gon Kim [1,2,6]

The cystic cavity that develops following injuries to brain or spinal cord is a major obstacle for tissue repair in central nervous system (CNS). Here we report that injection of imidazole-poly (organophosphazenes) (I-5), a hydrogel with thermosensitive sol–gel transition behavior, almost completely eliminates cystic cavities in a clinically relevant rat spinal cord injury model. Cystic cavities are bridged by fibronectin-rich extracellular matrix. The fibrotic extracellular matrix remodeling is mediated by matrix metalloproteinase-9 expressed in macrophages within the fibrotic extracellular matrix. A poly(organophosphazenes) hydrogel lacking the imidazole moiety, which physically interacts with macrophages via histamine receptors, exhibits substantially diminished bridging effects. I-5 injection improves coordinated locomotion, and this functional recovery is accompanied by preservation of myelinated white matter and motor neurons and an increase in axonal reinnervation of the lumbar motor neurons. Our study demonstrates that dynamic interactions between inflammatory cells and injectable biomaterials can induce beneficial extracellular matrix remodeling to stimulate tissue repair following CNS injuries.

[1] Department of Brain Science, Ajou University School of Medicine, 164 Worldcup-ro, Yeongtong-gu, Suwon 16499, Republic of Korea. [2] Neuroscience Graduate Program, Department of Biomedical Sciences, Ajou University Graduate School of Medicine, 164 Worldcup-ro, Yeongtong-gu, Suwon 16499, Republic of Korea. [3] Center for Biomaterials, Korea Institute of Science and Technology (KIST), 5 Hwarang-ro 14-gil, Seongbuk-gu, Seoul 02792, Republic of Korea. [4] Miami Project to Cure Paralysis, Department of Neurological Surgery, University of Miami School of Medicine, 1095 NW 14th Terrace (R-48), Miami, FL 33136, USA. [5] Department of Biomolecular Science, University of Science and Technology (UST), 217 Gajeong-ro, Yuseong-gu, Daejeon 34113, Republic of Korea. [6] Department of Neurology, Ajou University School of Medicine, 164, Worldcup-ro, Yeongtong-gu, Suwon 16499, Republic of Korea. Le Thi Anh Hong, Young-Min Kim and Hee Hwan Park contributed equally to this work. Correspondence and requests for materials should be addressed to S.-C.S. (email: scsong@kist.re.kr) or to B.G.K. (email: kimbg@ajou.ac.kr)

Traumatic or vascular injuries to the central nervous system (CNS) are frequently complicated by the development of fluid-filled cystic cavities[1–6]. A lack of extracellular matrix (ECM) and/or vascularization hinders infiltration of cellular elements and regeneration of axons in the cavity space[7]. Furthermore, survival and integration of transplanted cells for therapeutic purposes is substantially compromised in the presence of a cystic cavity[8, 9]. Therefore, the development of a cystic cavity poses a formidable hurdle for successful tissue repair after CNS injuries.

The fluid-filled cystic cavities are expected to have a particularly devastating influence after spinal cord injury (SCI) because the spinal cord is a cylindrical structure with a small cross-sectional area into which many important axonal tracts are crowded. Cystic lesions filled with cerebrospinal fluid are regularly present at the epicenter during the chronic stage of SCI[5]. Furthermore, more than 50% of patients develop posttraumatic spinal cord cysts or syringomyelia[10, 11]. Thus, it is important to target cystic cavities as part of any intervention for SCI.

Biomaterial-based treatment has been proposed as a strategy to promote tissue repair by bridging cavity spaces[12–15]. Implanting various tissue-engineered scaffolds or matrices has been reported to reduce cyst formation[16–19]. Since in most cases of human SCIs, the injuries are incomplete with a significant portion of the white matter spared[20], surgical procedures involving the implantation of scaffolds or matrices are prone to aggravating functional deficits. In this regard, there is a mounting consensus on the necessity of using injectable, in situ gelling hydrogels for future clinical translation[7, 14, 21]. However, whether injectable hydrogels can successfully induce bridging of posttraumatic cystic cavities in the spinal cord has not yet been convincingly demonstrated[22–26].

The present study reports that a polymer-based injectable hydrogel almost completely eliminates cystic cavities and leads to enhanced tissue repair in a clinically relevant contusive SCI model in rats. We found that the cystic spaces are bridged by fibronectin (FN)-rich ECM. Further experiments demonstrated that the hydrogel promotes ECM remodeling by activation of matrix metalloproteinase-9 (MMP-9) expressed in macrophages that are abundantly present in the newly remodeled fibrotic ECM. Our data also showed that the histamine moiety in the hydrogel and its physical interaction with macrophages plays an important role in the ECM remodeling. Thus, our study proposes that dynamic interaction between inflammatory cells in the host tissue and injectable hydrogel can be a principal mechanism underlying biomaterial-mediated tissue repair following CNS injuries.

## Results

**Synthesis and characterization of the polymer hydrogel.** Imidazole-poly(organophosphazenes) hydrogel was synthesized following the reaction schematic depicted in Fig. 1. We have previously synthesized polymer hydrogels with various functionalities derived from poly(organophosphazenes) (polymer I in Fig. 1)[27–29]. The poly(organophosphazenes) hydrogel is amphiphilic as it has both a hydrophobic isoleucine ethyl ester (IleOEt) and a hydrophilic α-amino-ω-methoxy-poly(ethylene glycol) (AMPEG), which impart temperature-dependent sol–gel transition behavior[30]. The hydroxyl group of 2-aminoethanol in poly(organophosphazenes) was esterified to provide a hydrolysable ester linkage and a terminal carboxylic acid group (polymer II in Fig. 1), which can be used to conjugate different functional moieties. In a pilot experiment aimed at searching for a candidate hydrogel suitable for bridging cystic cavities after SCI, we tested various conjugates based on the poly(organophosphazenes) polymer and found that imidazole-conjugated hydrogel showed striking effects. Therefore, we focused on the imidazole-poly(organophophazenes) polymer hydrogel (I-5) henceforth.

**Fig. 1** Schematic of the synthesis of I-5. *AMPEG*, α-amino-ω-methoxy-poly(ethylene glycol); *DMAP*, 4-(dimethylamino)pyridine; *DIC*, diisopropylcarbodiimide; *IleOEt*, isoleucin ethyl ester; *NHS*, N-hydroxysuccinimide; *THF*, tetrahydrofuran; *TEA*, trimethylamine

I-5 was synthesized through conjugation of imidazole to the polymer II (CP-2) by carbodiimide cross-linking between the amine group of 1–3 aminopropylimidazole and the carboxyl group of the polymer II (Fig. 1). The existence of the imidazole group in I-5 was confirmed by the presence of imidazole peaks in the $^1$H-NMR spectrum (Supplementary Fig. 1a) and by an increase of the C=O peak, corresponding to amide bond stretching in Fourier Transform InfraRed (FT-IR) spectroscopy data (Supplementary Fig. 1b). The molecular weight of I-5 ranged from approximately 14 kDa to 18 kDa The viscosity of I-5 hydrogel was measured at a wide range of temperatures (Fig. 2a). The viscosity rose abruptly when the temperature reached 30 °C, causing a phase transition from sol to gel state. The viscosity at body temperature was ~600 Pa·s, which indicates that the material injected in vivo has physical strength sufficient to support and maintain the shape of the hydrogel. The rapidity of the gelation process was visualized by injecting polymer solution into distilled water at either 4 or 37 °C (Fig. 2b, c). Immediately after injection of polymer solution into the water at 37 °C, rod-like gel formation was observed (Fig. 2c, *yellow dotted circle*). Several seconds later, opaque gel-like materials accumulated at the bottom of the glass vial (Fig. 2c, *white dotted rectangle*). In contrast, when the polymer solution was injected into 4 °C water, there was no evidence of gelation (Fig. 2b). We measured temporal changes in viscosity as a function of time elapsed after the temperature was set at 37 °C (Fig. 2d). Within 10 s after the temperature was fixed at 37 °C, hydrogel solution started to form gel-like material with a viscosity of ~50 Pa·s. The viscosity rose

very rapidly thereafter and reached a plateau approximately at 150 s, indicating completion of gelation within a few minutes at 37 °C. We also performed the in vitro stability test to obtain information on degradation behavior (Fig. 2e). In a solution set at 37 °C, I-5 hydrogel seemed to be swollen at 1 or 2 days in vitro, and then started to dissolve by 4 days but still persist at that time. By 7 days, gel mass disappeared and I-5 seemed to completely dissolve.

The viability of cultured cells was not affected by I-5 polymer solution at a wide range of concentrations (Supplementary Fig. 2a). When I-5 was subcutaneously injected into 6-week-old mice for in vivo biocompatibility testing, the mean body weight increased as expected (Supplementary Fig. 2b), suggesting that I-5 injection did not invoke systemic inflammatory reactions. Inspection of local injection sites also revealed no sign of inflammation or necrosis in the subcutaneous tissue surrounding the hydrogel (Supplementary Fig. 2c). Thus, I-5 polymer hydrogel is non-cytotoxic in vitro and compatible with the tissue environment not eliciting foreign body reactions in vivo.

**I-5 hydrogel eliminates cystic cavities after contusive SCI.** A previous study showed that cystic cavities start to form as early as 1 week after contusion injury and progressively enlarge up to the 4-week time point[31]. At 1 week after injury, we also observed quite large cystic cavities already formed at the epicenter (Supplementary Fig. 3a). At this time point, non-cystic lesions were filled with ED-1 positive macrophages surrounded

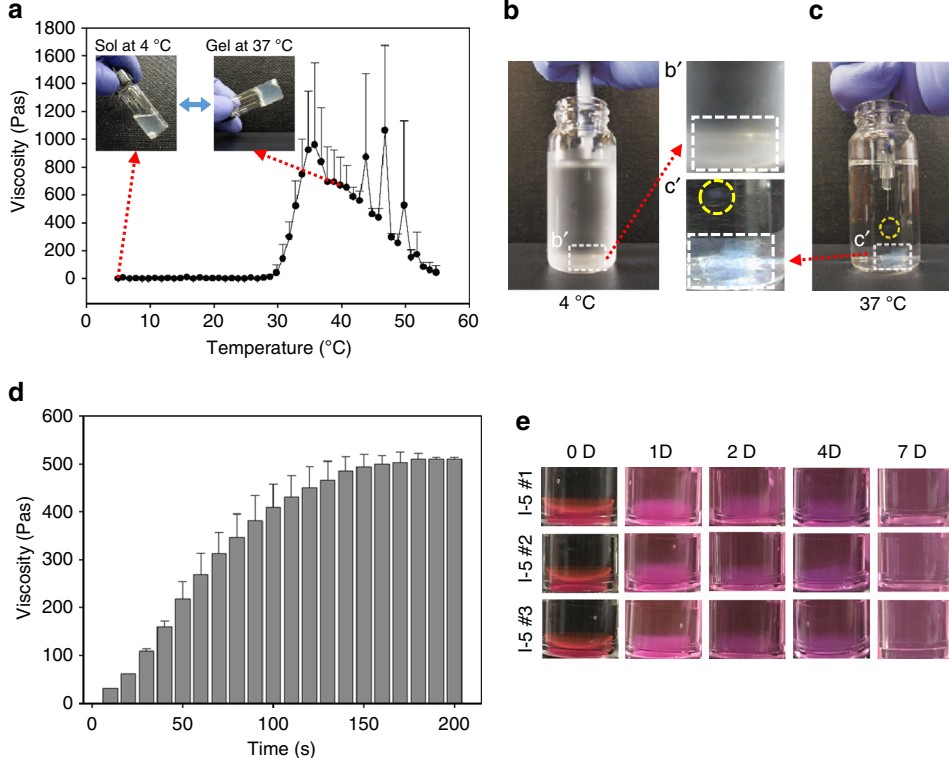

**Fig. 2** Sol–gel transition properties and in vitro stability test of I-5 hydrogel. **a** Temperature-dependent sol–gel transition and changes in viscosity measured by a viscometer. Each *black dot* represents the average value from four replicated measurements. *Error bars* represent standard deviation. **b**, **c** Visualization of the rapid gelation process. I-5 polymer solution was injected into a glass vial containing distilled water at either 4 °C (**b**) or 37 °C (**c**). After being injected into the water at 37 °C, the polymer solution rapidly turned into a rod-like gel material (*yellow dotted circle*). Gel-like materials accumulated at the bottom of the vial (*white dotted rectangle*, **c**). In comparison, there was no visible opaque gel-like material in the vial containing water at 4 °C (*white dotted rectangle*, **c**). Images in the white dotted rectangles in **b**, **c** were magnified in b', c'. **d** Changes in viscosity as a function of time elapsed after the temperature is set as 37 °C. Each bar is the average of three replicate measurements. *Error bars* represent standard deviation. **e** In vitro stability test. Three independent I-5 gels were monitored until 7 days in vitro at 37 °C

by glial fibrillary acidic protein (GFAP) positive astrocytes (Supplementary Fig. 3b). We chose to examine the presence of cystic cavities 4 weeks after injection because a previous study reported that there is no further expansion of the cystic cavities after the 4-week time point[31]. At 5 weeks after injury (4 weeks after injection), cystic cavities became larger extending rostrocaudally more than 2.0 mm away from the epicenter in animals injected with control PBS (Fig. 3a, c). Surprisingly, I-5 injection resulted in an almost complete disappearance of cystic cavities in all the animals that received I-5 injection (Fig. 3b, d). Instead of cystic cavities, the epicenter region was filled with eosin-stained ECM-like tissue (Fig. 3b). GFAP immunostaining showed that the ECM-like tissue was largely devoid of astrocytes, but surrounded by astrocytic scars (Fig. 3d). To quantitatively measure cavity volume, a series of cross-sectional images of the spinal cord were reconstructed to produce a 3D image (Fig. 3e; Supplementary Movie 1 and 2). As illustrated in the 3D images, the mean volume of cystic cavities was strikingly reduced in animals with I-5 injection (1.00 vs. 0.14 mm$^3$; $t_{(14)} = 4.292$, $p < 0.001$) (Fig. 3f). The volume of pathologic tissue, which was

defined as the area without normal tissue architecture, was significantly increased in animals with I-5 (Fig. 3g), probably due to the eosin-stained ECM-like tissue filling cystic spaces. I-5 injection significantly increased the volume of residual white matter (Fig. 3h). There were no detectable remnants of gel-like material within the spinal cord tissue. For an independent evaluation of the bridging effects by I-5, paraformaldehyde-fixed spinal cord tissues injected with PBS or I-5 were shipped to the laboratory of one of the coauthors (JKL), and the cavity formation was examined using a light sheet fluorescence microscope after tissue clearing, following the procedure published previously[32]. The 3D macroscopic imaging in whole spinal cord tissue also demonstrated a dramatic reduction of cystic cavities by I-5 injection (Supplementary Movies 3 and 4).

To examine the possibility that I-5 may provoke exaggerated inflammatory reactions in the spinal cord tissue, we examined the intensity of immunoreactivity for Iba1, a marker of inflammatory macrophages. At 1 week after injection, intense Iba1 immunoreactivities occupied at the lesion cores in animals with PBS or I-5 injection to a similar extent (Fig. 3i, j). At 4 weeks after

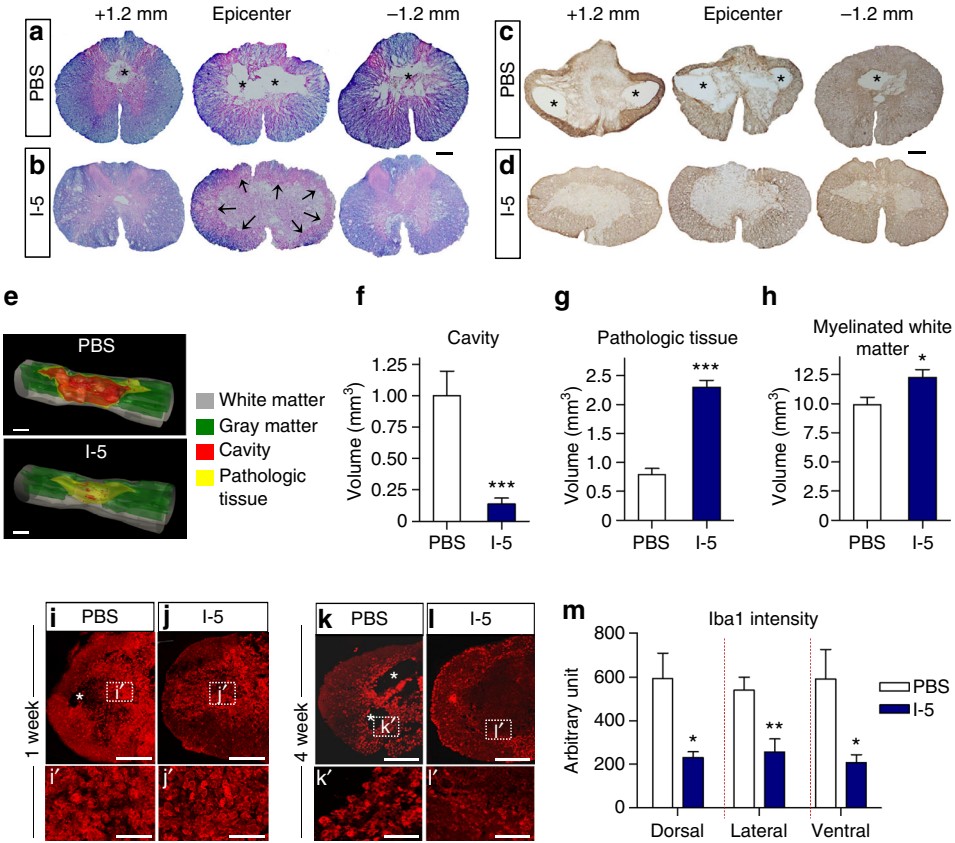

**Fig. 3** Injection of I-5 hydrogel eliminates cystic cavities. **a–d** Absence of cavity spaces after I-5 injection. Representative images of transverse spinal cord sections stained with eriochrome cyanine and eosin (**a, b**) or GFAP antibodies (**c, d**). Spinal cord sections were obtained from animals 4 weeks after PBS (**a, c**) or I-5 injection (**b, d**). The sections shown are from the epicenter and 1.2 mm rostral (+1.2 mm) or caudal (−1.2 mm) to it. *Asterisks* indicate cystic cavities. In animals with I-5 injection, there was no sizable cavity observed. Instead, eosin-stained ECM-like tissue was observed in the central epicenter region, of which boundary was indicated by black arrows (**b**). *Scale bars* represent 200 μm. **e** Three-dimensional reconstructions of the spinal cord tissues from animals with PBS or I-5 injection. *Scale bars* represent 1 mm. **f–h** Quantification graphs showing the volumes of cavity (**f**), pathologic tissue (**g**) and myelinated white matter (**h**). * and *** indicate $p < 0.05$ and $p < 0.001$, respectively, by two-tailed Student's *t*-test. $N = 8$ animals per group. *Error bars* represent the SEM. **i–l** Representative images of transverse spinal cord sections at the epicenter stained with an Iba1 antibody. The sections were obtained from animals 1 week (**i, j**) or 4 weeks (**k, l**) after injection. Magnified images of the boxed regions are shown in i′–l′. *Asterisks* indicate cystic cavities. *Scale bars* indicate 200 μm (**i–l**) and 50 μm (i′-l′). **m** Graph showing quantification of the Iba1 immunofluorescence intensity at 4 weeks after injection (5 weeks after injury). The fluorescence intensity was measured in three ROIs: one each in dorsal, lateral, and ventral regions containing the border between the residual white matter and damaged tissue with or without cavities at the epicenter. * and ** indicate $p < 0.05$ and $p < 0.01$, respectively, by two-tailed Student's *t*-test. $N = 6$ animals per group. *Error bars* represent the SEM

injection, strong Iba1 immunoreactivities remained surrounding cystic cavities in animals with PBS injection (Fig. 3k, m). In contrast, the intensity of Iba1 immunoreactivity was significantly attenuated at the border surrounding the ECM-like tissue filling the central region (Fig. 3l, m). These data suggest that I-5 did not provoke excessive inflammatory reactions and that bridging cystic cavities with I-5 may actually suppress post-injury inflammatory processes.

**I-5 induces formation of FN-rich fibrotic ECM**. Next, we characterized the ECM-like tissue at the epicenter region where cystic cavities are expected to be present without I-5 injection. In animals with PBS injection, GFAP-positive astroglia increased not only in the white matter but also in the central region of the injured tissue 1 week after injury (Fig. 4a). There were also areas devoid of GFAP immunoreactivity that were largely filled by FN immunoreactivity (Fig. 4a–c). In contrast, animals with I-5 injection showed more widespread formation of FN-positive matrix, whereas GFAP immunoreactivity was confined within the spared white matter (Fig. 4d). As a result, the FN-positive areas were largely separated from the GFAP-positive white matter although the two areas were not completely segregated (Fig. 4d–f). At 4 weeks after injury, cystic cavities were enlarged compared to the 1-week time point in animals with PBS injection (Fig. 4g). The expansion of cavity spaces was accompanied by shrinkage of the FN-positive fibrotic matrix (Fig. 4g–i). In contrast, the fibrotic matrix in animals with I-5 injection seemed to be consolidated at the central region: there was an increase in the density as well as intensity of FN immunoreactivity (Fig. 4j–k). The segregation between the FN- and GFAP-immunoreactive areas became more obvious than it was at the 1-week time point, establishing a discrete border between the two cellular compartments.

We noticed that the FN-rich fibrotic matrix formed by I-5 injection bore resemblance to the fibrotic scar observed after contusive SCI in mice, where cystic cavities do not develop[33]. Collagen 1α1-expressing perivascular fibroblasts are the major cellular source of the fibrotic scar in mice[33, 34]. Immunofluorescence staining against FN and collagen 1α1 showed colocalization within the FN-rich ECM formed by I-5 injection (Supplementary Fig. 4a, b). Cells that are positive for PDGFR-beta, a marker for perivascular fibroblasts[33], were frequently found within the matrix encircling RECA-1 immunoreactive endothelial cells (Supplementary Fig. 4c). It was reported that macrophages mediate assembly of FN matrix in the fibrotic scar in mice[34]. We found that the FN-rich ECM was densely populated by ED1- and CD11b-positive macrophages at 1 and 4 weeks after I-5 injection (Supplementary Fig. 4d, e). These macrophages were highly likely of a hematogenous origin because CD45 immunoreactivity was also observed at the 1-week time point (Supplementary Fig. 4d)[35]. The intensity of CD45 immunoreactivity tended to decline but still clearly persisted at the 4-week time point (Supplementary Fig. 4e). Those macrophages were also positive with CD206, a maker of M2 polarization[36] (Supplementary Fig. 4d, e).

The above findings suggest that fibroblasts and fibrotic scar formation may play a critical role in the hydrogel-induced elimination of cystic cavities. A previous study demonstrated that administration of the microtubule stabilizer Taxol can decrease fibrotic scar formation[37]. To examine a potential causative role of fibrotic scarring in the bridging effects, I-5 hydrogel mixed with Taxol or PBS was injected. As expected, I-5 mixed with PBS prevented development of cystic cavities by inducing FN-rich ECM formation (Fig. 5a–c). Injection of I-5 mixed with Taxol resulted in a reduction of FN-positive ECM and marked expansion of cystic cavities (Fig. 5d–f, g).

**MMP-9 in macrophages mediates ECM remodeling**. We speculated that certain matrix remodeling enzymes may be involved in the formation of FN-rich fibrotic matrix after I-5 injection. MMPs are zinc-dependent endopeptidases capable of modulating ECM proteins[38] and MMPs with gelatinase activity have beneficial roles in matrix remodeling and wound healing-associated fibrosis[39, 40]. In animals with hydrogel injection, the activity of MMP-9, but not MMP-2, was noticeably enhanced in zymography (Fig. 6a–c). Immunostaining to detect MMP-9 revealed very little immunoreactivity in the remaining matrix surrounding cystic cavities (Fig. 6d). In contrast, there was

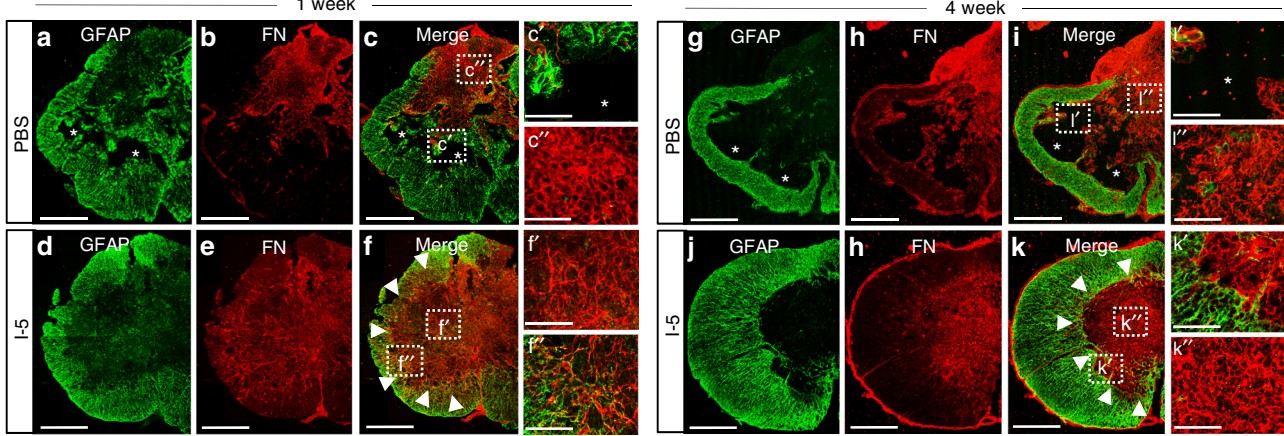

**Fig. 4** Remodeling of extracellular matrix by I-5 injection. **a–f** Representative images of transverse spinal cord sections obtained at 1 week after injection from animals with PBS (**a–c**) or I-5 (**d–f**) injection. Arrowheads indicate the border between the GFAP- and FN-positive areas. Boxed regions in **c**, **f** were magnified in c′, c″, f′, f″. **c′** Early cavity space (*) surrounded by GFAP-immunoreactive astroglial cells. **c″** FN-rich extracellular matrix (*ECM*). **f′** FN-rich ECM. **f″** GFAP and FN immunoreactivity intermingled in the border region. **g–k** Representative images of transverse spinal cord sections obtained 4 weeks after injection from animals with PBS (**g–i**) or I-5 (**j–k**) injection. Arrowheads indicate the border between the FN and GFAP immunoreactivities. Boxed regions in **i**, **k** were magnified in i′, i″, k′, k″. **i′** A cystic space (*) with residual FN-immunoreactive matrix. **i″** FN-rich ECM at the periphery of the lesion center. **k′** The sharp boundary between the FN- and GFAP-positive areas. **k″** Formation of dense and strong FN-rich matrix in the center of the lesion. All *asterisks* indicate cystic cavities. *Scale bars* represent 500 μm in **a–k**. Scale bars in the magnified images represent 100 μm

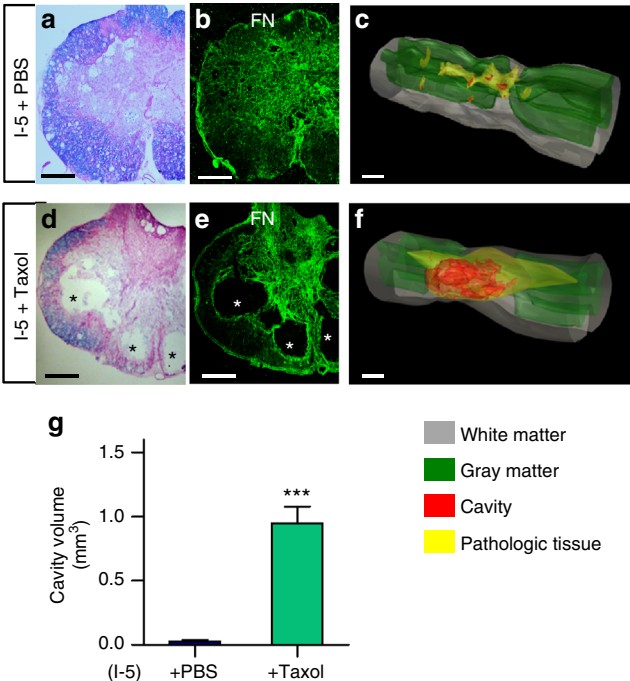

**Fig. 5** Taxol results in expansion of cystic cavities. **a, d** Representative images of transverse spinal cord sections stained with eriochrome cyanine and eosin from animals with I-5 mixed with PBS (**a**) or Taxol (**d**). Adjacent sections were stained with antibodies against FN (**b, e**). *Asterisks* indicate cystic cavities. *Scale bars* represent 200 μm. **c, f** Representative 3D reconstruction images of the spinal cord tissue from animals with I-5 mixed with PBS (**c**) or Taxol (**f**). *Scale bars* represent 1 mm. **g** Quantification graph of cavity volume. *** indicates $p < 0.001$ by two-tailed Student's *t*-test. $N = 4$ for each group. *Error bars* represent the SEM

markedly increased expression of MMP-9 within the FN-rich fibrotic matrix in hydrogel-injected animals (Fig. 6e). Interestingly, the MMP-9 immunoreactivity appeared granular and was largley surrounded by immunoreactivity for the macrophage cell surface marker CD11b (Fig. 6f).

To determine whether MMP-9 mediates the fibrotic ECM remodeling induced by I-5 hydrogel, we performed a knockdown experiment using MMP-9 siRNA delivered by nanoparticles. Suppression of MMP-9 expression by siRNA nanoparticles was validated in cultured peritoneal macrophages (Supplementary Fig. 5a, b). Injection of I-5 hydrogel mixed with MMP-9 siRNA nanoparticles, but not injection of I-5 mixed with nanoparticles carrying non-targeting control siRNA, depleted MMP-9 immunoreactivity in the FN-positive ECM (Supplementary Fig. 5d, e). MMP-9 knockdown decreased formation of FN-rich matrix at the lesion epicenter and substantially enlarged cystic cavities (Fig. 6j–l, m). In contrast, injection of I-5 with non-targeting siRNA almost completely eliminated cystic spaces (Fig. 6g–i, m). These findings indicate that macrophages within the FN-rich matrix newly formed after I-5 hydrogel injection may produce MMP-9 enzymes that promote remodeling of fibrotic ECM at the epicenter.

**Interaction between macrophages and I-5 hydrogel.** Recent studies have shown that in the injured mouse spinal cord, hematogenous macrophages are closely associated with collagen 1α1-positive perivascular fibroblasts and play a role in recruiting fibroblasts and promoting FN matrix assembly[34, 41]. Therefore, our data from the above experiments collectively suggested that

macrophages found within the FN-rich matrix might drive recruitment of perivascular fibroblasts and remodeling of ECM leading to formation of the fibrotic matrix. I-5 hydrogel contains an imidazole group, a main moiety of histamine, and macrophages express two types of histamine receptor, $H_1R$ and $H_4R$[42]. Therefore, we tested if I-5 hydrogel physically interacts with macrophages through binding to their histamine receptors. To visualize the interaction between the polymer hydrogel and macrophages in vitro, poly(organophosphazenes) polymer solutions were mixed with the hydrophobic fluorescent dye Nile Red. The hydrophobic interaction between the polymer and Nile Red induced the formation of nano-scaled polymer micelles emitting red fluorescence. We expected that if the macrophages physically interact with the polymer hydrogels, hydrophobic Nile Red dye in the nanoparticles would be delivered into macrophages. RAW 264.7 mouse macrophages were incubated with the nanoparticles consisting of Nile Red and I-5 polymer micelles containing the imidazole group or CP-2 (carboxylic acid terminated polymer II in Fig. 1) without the imidazole group. When Nile Red dye alone was added to the culture medium, very little fluorescence was emitted from macrophages (Fig. 7a). Macrophages exhibited discernible red fluorescence after incubation with the nanoparticles composed of CP-2. However, the fluorescence intensity was markedly higher when treated with the I-5 nanoparticles (Fig. 7a, b), demonstrating that the presence of the imidazole group enhances interaction between the polymer hydrogel and macrophages. The red fluorescence in macrophages was lower after pretreatment with JNJ777120 or mepyramine, $H_4R$ and $H_1R$ inhibitors, respectively (Fig. 7a, b), suggesting that the imidazole group and histamine receptors in macrophages mediate the interaction between the polymer and macrophages.

If the interaction between the I-5 hydrogel and macrophages significantly contributes to its bridging effect, injection of poly (organophosphazenes) hydrogel without an imidazole ring structure would be predicted to result in expansion of cystic cavities. We injected CP-2 hydrogel lacking the imidazole moiety and compared its effects to those of I-5. The viscosity at 37 °C of CP-2 was close to 600 Pa·s that was very similar to that of I-5 (Supplementary Fig. 6a). Furthermore, in vitro stability test showed dissolution of CP-2 occurring also between 4 and 7 days (Supplementary Fig. 6b), indicating that the physical properties of the two hydrogels are very similar to each other. Unlike I-5, however, injection of CP-2 lacking the imidazole moiety failed to eliminate large cystic cavities (Fig. 7c–f).

**I-5 injection promotes functional recovery and tissue repair.** To assess functional recovery after injection with I-5 or PBS, Basso, Beattie and Bresnahan (BBB) open field locomotor scores were determined during the 8 weeks after injury (7 weeks after PBS or I-5 injection). Animals in both groups showed spontaneous recovery over that time frame (Fig. 8a). However, animals with I-5 injection showed enhanced recovery from the 4-week time point and thereafter. Repeated measures two-way ANOVA revealed a significant influence of I-5 treatment on behavioral recovery ($F_{(1, 120)} = 5.265$, $p < 0.05$), and the interaction between treatment and time points was also significant ($F_{(8, 120)} = 3.738$, $p < 0.001$). We tested locomotor recovery further by Catwalk analysis (Fig. 8b). Mean stride length was not significantly different between the treatment groups (including the sham-operated group) (Fig. 8c). Neither the base of support (the width between the left and right hindpaws) nor the angle of hindpaw rotation was improved after I-5 injection (Fig. 8d, e). The engagement of fore- and hindpaws during locomotion became uncoordinated after injury, resulting in non-overlapping fore- and hindpaws footprints (Fig. 8b) and an

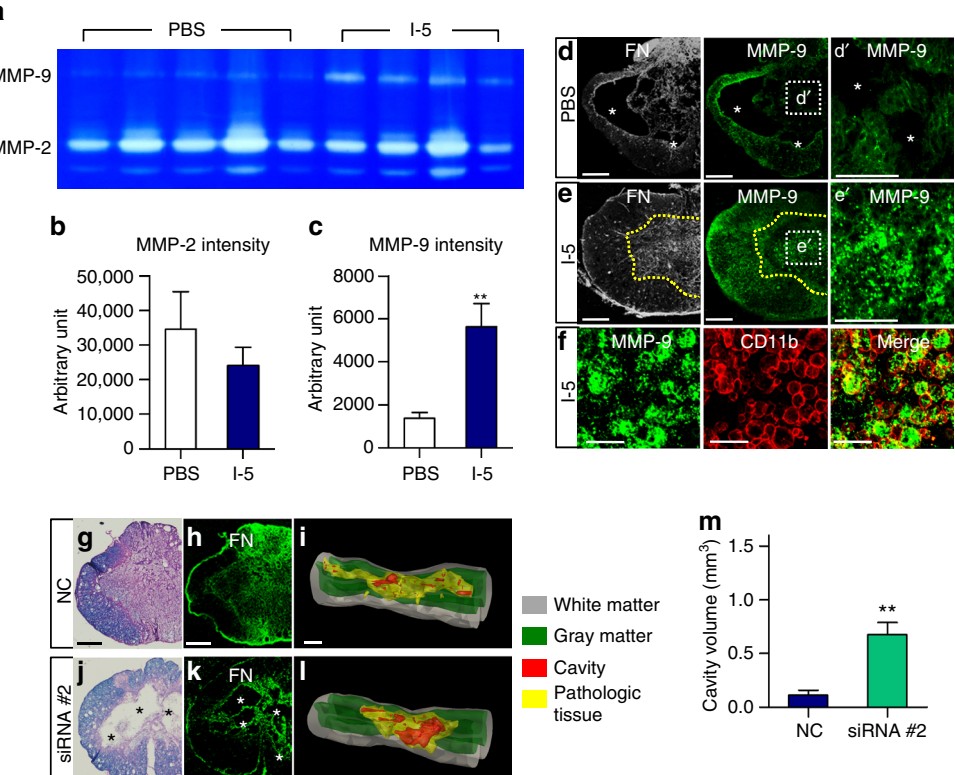

**Fig. 6** MMP-9 mediates fibrotic ECM remodeling. **a** Gelatinase activity of MMP-9 and MMP-2 in the PBS ($N = 5$) and the I-5 group ($N = 4$) examined by zymography. **b**, **c** Graphs showing quantification of MMP-2 (**b**) and MMP-9 (**c**) activity. ** indicates $p < 0.01$ by two-tailed Student's $t$-test. *Error bars* represent the SEM. **d**, **e** Representative images of transverse spinal cord sections from animals injected with either PBS (**d**) or I-5 (**e**). *Asterisks* indicate cystic cavities. A *yellow dotted line* indicates the boundary of the FN-rich matrix. The boxed regions in the middle panels were magnified in d′, e′. *Scale bars* represent 100 μm. **f** MMP-9 immunoreactive granules were bounded by CD11b positive membrane-like circular structures. *Scale bars* represent 50 μm. **g**, **j** Representative images of transverse spinal cord sections stained with eriochrome cyanine and eosin from animals with I-5 mixed with non-targeting control siRNA (NC) (**g**) or MMP-9 siRNA (**j**). Adjacent sections were stained with antibodies against FN (**h**, **k**). *Scale bars* represent 200 μm. **i**, **l** 3D reconstruction of the spinal cord tissue using the Neurolucida software. *Scale bars* represent 1 mm. *Asterisks* indicate cystic cavities. **m** Quantification graph of cavity volume. ** indicates $p < 0.01$ by two-tailed Student's $t$-test. $N = 5$ for each group. *Error bars* represent the SEM

increase in relative position (the distance between the ipsilateral fore- and hindpaws in one step cycle) (Fig. 8f). I-5 injection significantly reduced relative position, suggesting that injection of I-5 resulted in enhanced coordination between the fore- and hindpaws. The regularity index, which was developed to determine coordinated gait[43], was significantly reduced after injury, and I-5 injection tended to increase the regularity index (Fig. 8g).

To investigate potential mechanisms by which I-5 injection enhances functional outcomes, the number of surviving motor neurons in the ventral horns caudal to the injury epicenter was analyzed 8 weeks after injury (Fig. 9a–c). Compared to the motor neurons in sham-operated animals, surviving neurons in the ventral horns were very rarely observed 1.2 mm caudal to the lesion in either group (Fig. 9b, c). At 1.6 mm caudal to the lesion, several motor neurons per section were observed with a tendency for the number to be higher in the I-5 injection group. The difference became statistically significant 2.0 mm caudal to the lesion where an average of more than 10 surviving neurons were found in the I-5 injection group (Fig. 9g). We also noticed that myelin basic protein (MBP) immunoreactive signal intensity was significantly reduced in the white matter after injury, particularly in the white matter surrounding cystic cavities (Fig. 9d, e). In animals with I-5 injection, the MBP signal intensity was substantially increased (Fig. 9f). Quantification data showed that I-5 significantly increased the MBP immunoreactivity at the epicenter (Fig. 9i). I-5 injection also tended to increase the MBP

signal at both 2.0 mm rostral and caudal to the epicenter, especially in the ventral region (Fig. 9h, j).

We also examined if regenerating axons could grow into the FN-rich ECM. A substantial number of the neurofilament (NF)-positive axons were frequently observed within the newly formed fibrotic matrix (Fig. 10a). Serotonergic (5-HT) axons, which may play a role in locomotor recovery, also grew into the FN-rich matrix (Fig. 10b). We also performed antergrade tracing. First, AAV8-GFP was injected into the sensorimotor cortex to visualize the corticospinal axon. GFP positive corticospinal axons were nicely visualized up to several millimeters rostral to the epicenter, but then sudden stopped growing (Fig. 10c). There was no discernible GFP positive axonal fibers within the FN-rich matrix (marked by the *yellow dashed line*). Next, biotinylated dextran amine (BDA) was injected into upper thoracic spinal cord to label axons various descending axons (either supraspinal or long propriospinal). Quite a large extent of BDA-traced axons grew beyond the rostral border of the FN-rich matrix (Fig. 10c). The amount of axons visualized decreased in deeper areas farther from the border as the intensity of FN immunoreactivity increased. Overt axonal fibers were occasionally observed in the central region of FN-rich matrix. However, there was no BDA-traced axons regnerating beyond the caudal border of the FN-rich matrix. Finally, we evaluated the extent of 5-HT axon innervation in the ventral motor regions of the lumbar spinal cord. At 1 week following injury, the density of 5-HT axons in the ventral horn of the lumbar spinal cord was sharply reduced

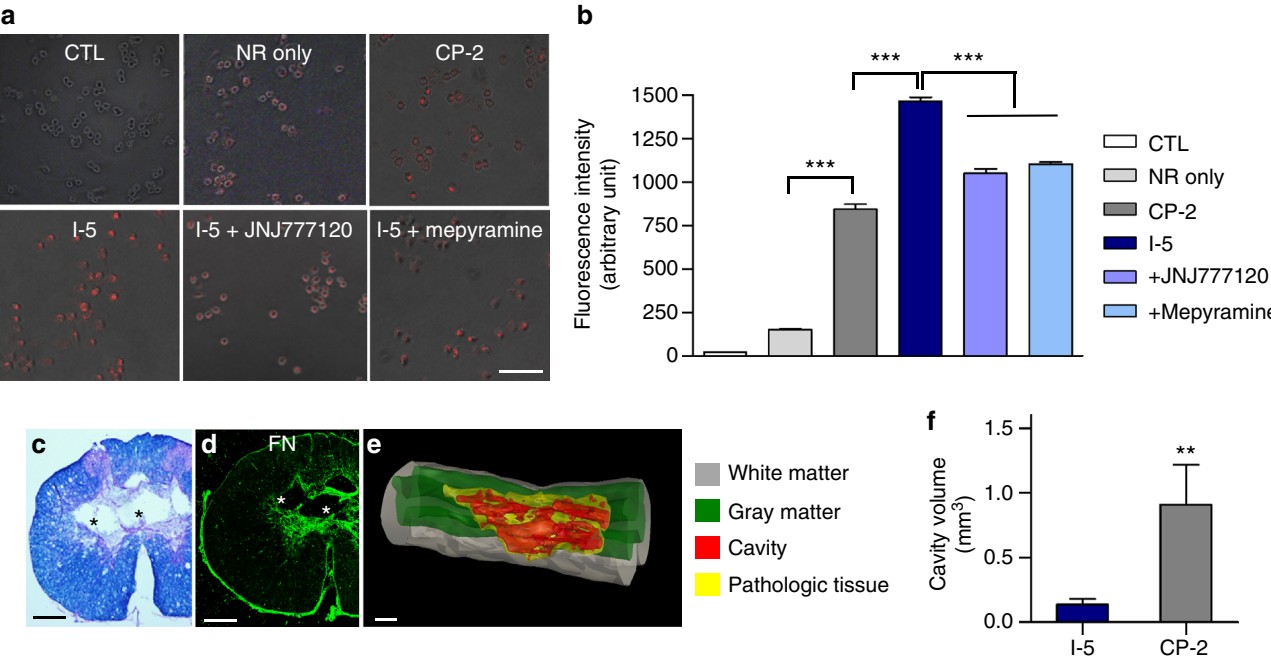

**Fig. 7** Interaction between macrophages and I-5 hydrogel. **a** Representative images of Nile Red fluorescence in a macrophage cell line. Cultured macrophages were treated with Nile Red (NR) alone or nanoparticles consisting of NR and either CP-2 or I-5. In addition, JNJ7777120, a histamine receptor 4 inhibitor, or mepyramine maleate, a histamine receptor 1 inhibitor, was added to the culture medium 30 min before treatment with I-5 polymer nanoparticles. *Scale bar* represents 50 μm. **b** Graph showing quantification of NR fluorescence intensity. *** indicates $p < 0.001$ by one-way ANOVA followed by Tukey's post hoc analysis. $N = 4$ replicate experiments per group. *Error bars* represent the SEM. **c**, **d** Representative images of transverse spinal cord sections from animals injected with CP-2 hydrogel lacking the imidazole group. **c** Eriochrome cyanine and eosin staining revealed prominent cystic cavities (*) at the center of the lesion (**c**). **d** FN staining showed a smaller area of FN-rich matrix. *Scale bars* represent 200 μm. **e** 3D reconstruction of the spinal cord tissue from an animal injected with CP-2 hydrogel using the Neurolucida software. *Scale bar* represents 1 mm. **f** Graph showing the quantification of the cavity volumes. The data set for the I-5 group was the same as the one used in Fig. 3f. ** indicates $p < 0.01$ by two-tailed Student's *t*-test. $N = 8$ for the I-5 group and $N = 5$ for the CP-2 group. *Error bars* represent the SEM

compared to the sham-opreated animals (Fig. 10c, d). The 5-HT axon density was not signficantly changed for the next 7 days by injection of either PBS or I-5. However, the extent of 5-HT innervation increased substantially at the 8-week time point in animals with I-5 injection, but not those with PBS control (Fig. 10e). These results suggest that the FN-rich matrix formed by I-5 injection may promote axonal reinnervation of the lumbar motor regions and thereby contribute to the recovery of locmotor function.

## Discussion

Cystic cavities in the majority of human SCIs have highly irregular and unpredictable geometry. However, many studies tested injectable biomaterials in surgically created injury models that have a predetermined dimension of the injury site[44–48], making it difficult to evaluate their potential as a clinically applicable bridging strategy. Our study demonstrated that injection of I-5 hydrogel can induce near complete bridging of posttraumatic cavities using a clinically relevant contusion injury model in rats, where fluid-filled cystic cavities progressively develop in a manner closely resembling what occurs after human SCI[31, 49].

To the authors' knowledge, the extent of the bridging effects in the current study has hitherto not been reported with other types of injectable matrices used in clinically relevant injury models similar to ours. In previous reports, injectable self-assembling matrices were reported to support axon growth within and around the cystic lesions in contusive SCI models, but did not reduce cavity size[23, 25]. Injection of different types of self-assembling peptides or hyaluronan-based hydrogels reduced

cavity volume or promoted tissue preservation following clip compression injury, but sizable cystic cavities remained[22, 24, 26]. In stark contrast to these reports, our I-5 hydrogel almost completely eliminated cavity spaces. Our data also provide evidence that I-5 hydrogel interacts with macrophages that are activated to produce MMP-9, which in turn promotes fibrotic ECM remodeling. Therefore, the superior bridging effects of I-5 hydrogel likely result from its dynamic interaction with cellular components and/or interstitial matrix in the host tissue in addition to its primary function of physical support or structural stabilization.

Injection of I-5 mixed with Taxol, previously shown to reduce fibrotic scarring[37, 50], resulted in the reduction of FN-rich ECM and expansion of cavity spaces. These data strongly suggest that the mechanism of FN-rich matrix formation is similar to that of fibrotic scarring and is critically involved in I-5 hydrogel-induced tissue repair. Intriguingly, the fibrotic matrix surrounded by GFAP-positive astrocytes in rats with I-5 injection in our study seems to have a close similarity to the fibrotic scars described in mouse contusive SCI[33]. In contrast to rats and humans, the mouse spinal cord does not develop cystic cavities after contusive or compressive injuries but becomes filled with dense fibrotic connective tissue[51, 52]. Thus, it is conceivable that a potential mechanism by which I-5 injection promotes ECM remodeling in the rat spinal cord may be similar to that occurring in fibrotic scar formation in the mouse spinal cord. Indeed, cell type analysis in the newly formed ECM revealed that perivascular fibroblasts, the main cellular source of the fibrotic matrix in mice[33], also constitute the major cellular component in the fibrotic ECM formed by I-5 injection in rats.

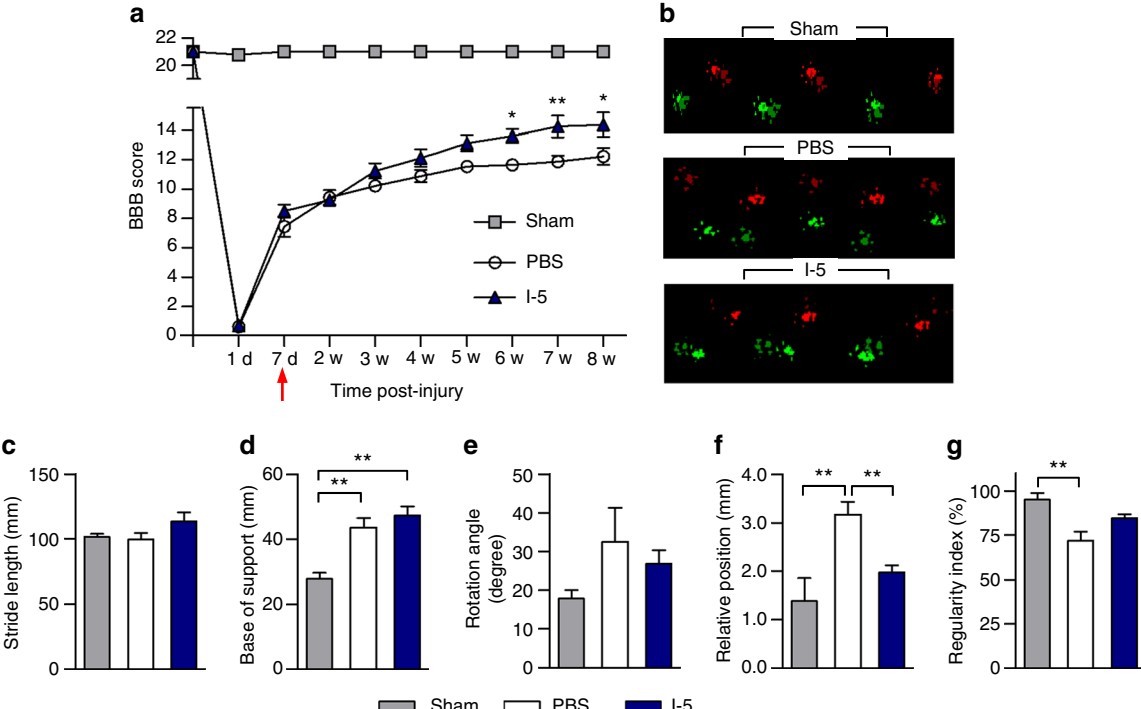

**Fig. 8** Injection of I-5 hydrogel promotes locomotor recovery. **a** Comparison of locomotor recovery in the PBS and I-5 injection groups. Locomotor recovery was measured by using the BBB scale in an open field. Animals with I-5 injection showed improved BBB locomotor scores from 4 weeks after injury. A red arrow indicates the day of injection. * and ** Indicate $p < 0.05$ and $p < 0.01$, respectively, by repeated measures two-way ANOVA followed by post hoc Bonferroni analysis. $N = 9$ for the PBS group and $N = 8$ for the I-5 groups. *Error bars* represent the SEM. **b** Representative footprints analyzed by the Catwalk software. The colors of the footprints were assigned by the Catwalk software (*bright red* = left forepaw, *dim red* = left hindpaw, *bright green* = right forepaw, and *dim green* = right hindpaw). **c–g** Five parameters were used to quantify the quality of locomotion 7 weeks after injury. **c** Stride length. **d** Base of support. **e** Rotation angle. **f** Relative position. **g** Regularity index. ** indicates $p < 0.01$ by one-way ANOVA followed by Tukey's post hoc analysis. $N = 5$, 9 and 8 for the sham, PBS, and I-5 groups, respectively. *Error bars* represent the SEM

The abundant presence of macrophages within the fibrotic matrix suggested their potential interaction with I-5 hydrogel. Our data indicated that the imidazole ring structure of I-5 may be responsible for its interaction with histamine receptors on macrophages, and that the interaction between I-5 and macrophages may trigger fibrotic ECM remodeling leading to bridging of cystic cavities. Macrophages are the key cells that orchestrate wound healing processes to repair tissue defects[53]. Therefore, it is conceivable that I-5 hydrogel may maintain the presence of macrophages at the lesion epicenter through interaction between the imidazole moiety and histamine receptors and that the prolonged presence of macrophages may lead to activation of wound healing mechanisms. The presence of macrophages positive with CD206, a marker of M2 polarization, supports this notion. Macrophages can also recruit perivascular fibroblasts and trigger assembly of FN matrix[34, 41]. We provide evidence that MMP-9, which was highly upregulated in macrophages within the fibrotic matrix, may play a critical role in the matrix remodeling process. Interestingly, FN matrix can also stimulate MMP-9 expression[54, 55]. Therefore, macrophages stabilized at the lesion site by I-5 hydrogel may play a central role in the complex communications between MMP-9, perivascular fibroblasts, and FN matrix. Our in vitro stability test suggests that the I-5 hydrogel would be degraded within one week and it is highly likely that in vivo degradation would be faster. We propose that the major function of I-5 hydrogel in our model was to trigger dynamic interaction with macrophages early after injection and thereby activate macrophage-mediated wound healing responses and fibrotic ECM remodeling in the ensuing period.

Several structural changes may explain how neural function for locomotion was recovered by I-5 injection. We found that filling cavity spaces with fibrotic matrix resulted in preservation of motor neurons and myelination. In animals with control PBS injection, myelin immunoreactivity was severely reduced especially in the spared white matter surrounding cavity spaces. We speculate that fluid within the cystic spaces may contain elevated levels of toxic substances that can contribute to secondary degeneration of motor neurons and/or demyelination of the white matter. Filling cystic spaces with fibrotic matrix may minimize exposure to toxic materials in the cystic fluid, leading to preservation of myelin and motor neurons. Another possibility is that the formation of fibrotic matrix in the lesion center may increase the stability of post-injury spinal cord. Spinal cord tissue with a large fluid-filled cavity would be more vulnerable to physical alterations such as tissue collapse or adhesion to bones and tendons, which could result in functional exacerbation. Presence of NF positive and 5-HT axons within the FN-rich matrix implies that the newly remodeled fibrotic matrix is permissive for axon growth. BDA-traced descending axons also entered into the matrix. Furthermore, we found an increase in 5-HT axon density in the lumbar spinal cord by I-5 injection, which may have direct relevance with the locomotor recovery. Innervation of 5-HT axons up to 2 weeks after injury was markedly reduced, correlating with the locomotor deficits at this time point. The substantial increase of 5-HT axons between the 2- and 8-week time points suggests that the FN matrix formation by I-5 injection supports reinnervation rather than sparing of existing axons in the ventral motor regions at the lumbar level. Previous studies suggested that fibrotic scar formation could be

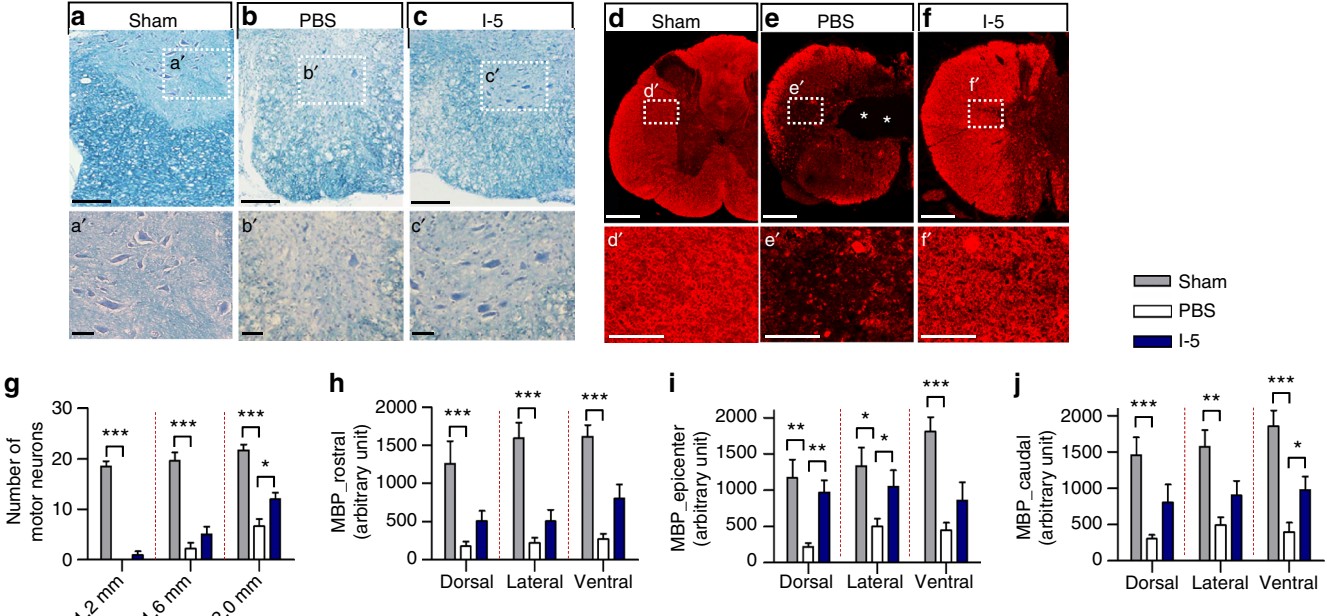

**Fig. 9** I-5 injection preserves motor neurons and myelinated white matter. **a–c** Representative images of the ventromedial region in transverse spinal cord sections 1.6 mm caudal to the epicenter from animals in sham-operated (**a**), PBS (**b**) and I-5 (**c**) injection groups. The sections were stained with eriochrome cyanine and cresyl violet. Boxed regions in **a–c** are magnified in a′–c′. Surviving motor neurons were more frequently observed in the I-5 than PBS injection group. *Scale bars* represent 200 μm in **a–c** and 50 μm in a′–c′. **d–f** Representative images of MBP immunostaining in transverse spinal cord sections of the lesion epicenter from animals in sham-operated (**d**), PBS (**e**) and I-5 (**f**) groups. Boxed regions in **d–f** are magnified in d′–f′. *Scale bars* represent 500 μm in **d–f** and 50 μm in d′–f′. **g** Quantitative comparison of the number of motor neurons. **h–j** Graphs showing quantification of the fluorescence intensity of MBP immunoreactivity. Quantification was performed in the sections located 2 mm rostral (**h**) to the epicenter, at the epicenter (**i**), and 2 mm caudal (**j**) to the epicenter. At each level, three ROIs were selected: one each in dorsal, lateral, and ventral white matter. *, ** and *** indicate $p < 0.05$, $p < 0.01$ and $p < 0.001$, respectively, by one-way ANOVA followed by Tukey's post hoc analysis. $N = 5$, 9 and 8 for the sham-operate, PBS and I-5 groups, respectively. *Error bars* represent the SEM

detrimental to the repair of injured CNS tissue by inhibiting axon regeneration[6, 37]. On the other hand, FN is a well-known substrate for cell adhesion, and previous studies have shown that FN matrix can support axon growth in in vivo SCI model[18, 47]. Moreover, FN can provide neuroprotective effects via multiple signaling pathways[56, 57]. Therefore, the formation of fibrotic ECM, which shares a similar mechanism to fibrotic scarring, may contribute to beneficial effects of I-5 hydrogel in tissue preservation and axonal growth.

To summarize, we show that I-5 hydrogel induces dramatic bridging effects on posttraumatic cavities and enhances tissue repair by promoting ECM remodeling. The rapid degradation of the hydrogel without invoking inflammatory reactions within the host tissue adds more merit to using this hydrogel as an injectable biomaterial. Furthermore, our study has already demonstrated the versatility of this hydrogel as a carrier of therapeutic reagents: I-5 successfully delivered a drug (Taxol) and nucleotides (siRNA) in our experiments. One of the advantages of polymer-based synthetic hydrogel is the ease of their synthetic modification, allowing engineered design for improved functionality[21]. Designing functional moieties conjugated to the poly(organo-phosphazenes) should allow for flexible control over the nature and/or strength of the interaction with inflammatory cells or even other cellular components in the host tissue to fine-tune the hydrogel into an optimal biomaterial for SCI repair.

## Methods

**Synthesis of the imidazole-poly(organophosphazenes) hydrogel**. All reactions were proceeded under a dry nitrogen atmosphere using standard Schlenk-line techniques.

Step 1: $[NP(IleOEt)_{1.45}(AEtOH)_{0.14}(AMPEG750)_{0.41}]_n$ (Polymer I, EP)

IleOEt·HCl (23.81 g, 121.67 mmol) in dry tetrahydrofuran (THF) containing trimethylamine (TEA) was added slowly to poly(dichlorophosphazene) (10.00 g, 86.29 mmol) dissolved in dry THF. The reaction mixture was incubated at dry ice bath for 12 h and room temperature for 36 h. AEtOH (1.56 g, 25.50 mmol) and AMPEG750 (37.53 g, 50.04 mmol) were dissolved in dry THF including TEA and added to the mixture. The reaction mixture was stirred at room temperature for 24 h and then at 40–50 °C for 24 h. AMPEG750 (18.76 g, 25.02 mmol) in dried THF was added to the reaction mixture and stirred at room temperature for 24 h and then at 40–50 °C for 24 h. The reaction mixture was filtered; the filtrate was concentrated and poured into n-hexane to obtain a precipitate, which was re-precipitated twice in the same solvent system. The polymer product was further purified by dialysis with a dialysis membrane (Spectra/Por, MWCO: 10–12 kDa) against methanol for 4 days at room temperature and against distilled water for 4 days at 4 °C. The dialyzed solution was freeze-dried to obtain polymer I. Yield: 82%. $^1$H NMR (CDCl$_3$), $\delta$ (ppm): 0.8–1.0 (s, 6H), 1.1–1.3 (b, 3H), 1.3–1.6 (b, 2H), 1.6–1.9 (b, 1H), 2.8–3.3 (b, 2H), 3.4–3.8 (b, 73H), 3.9 (s, 1H), 4.0–4.3 (b, 3H).

Step 2: $[NP(IleOEt)_{1.45}(Succinic\ acid)_{0.14}(AMPEG750)_{0.41}]_n$ (Polymer II, CP-2). Succinic anhydride (2.76 g, 27.58 mmol) and 4-(dimethylamino)pyridine (DMAP) (3.37 g, 27.56 mmol) dissolved in dried THF, were added separately into EP (27.12 g, 45.94 mmol) in dried THF. The reaction mixture was stirred at 40 °C for 24 h. The products were dialyzed with a dialysis membrane (Spectra/Por, MWCO: 10–12 kDa) against methanol for 4 days at room temperature and against distilled water for 4 days at 4 °C. Freeze-drying was carried out after dialysis to obtain polymer II. Yield: 91%. $^1$H NMR (CDCl$_3$), $\delta$ (ppm): 0.8–1.0 (s, 6H), 1.1–1.3 (b, 3H), 1.3–1.6 (b, 2H), 1.6–1.9 (b, 1H), 2.5–2.8 (b, 2H), 2.8–3.3 (b, 2H), 3.4–3.8 (b, 62H), 3.9 (s, 1H), 4.0–4.3 (b, 3H).

Step 3: $[NP(IleOEt)_{1.45}(Succinic\ acid)_{0.13}(Imi)_{0.01}(AMPEG750)_{0.41}]_n$ (I-5). The carboxylic acid terminated polymer II (10 g, 16.55 mmol) was dissolved in dried THF. After cooling to 0 °C, diisopropylcarbodiimide (DIC; 3.58 ml, 23.17 mmol) and N-hydroxysuccinimide (NHS; 2.67 g, 23.16 mmol) were added and the mixture was stirred for 30 min in order to activate the carboxyl groups of the polymer. The mixture was moved to 1–3-aminopropylimidazole (1.45 g, 11.54 mmol) dissolved in distilled THF. The reaction mixture was stirred at 0 °C for 6 h and then at room temperature for 18 h. Afterwards, the solution was concentrated and purified by precipitation with 1 M potassium fluoride solution. The precipitate was dialyzed with a dialysis membrane (Spectra/Por, Spectrum Laboratories, molecular weight cut-off: 12–14 kDa) against distilled water for 3 days at 4 °C and the dialyzed solution was freeze-dried to obtain the final product (I-5). Yield: 79%. $^1$H NMR

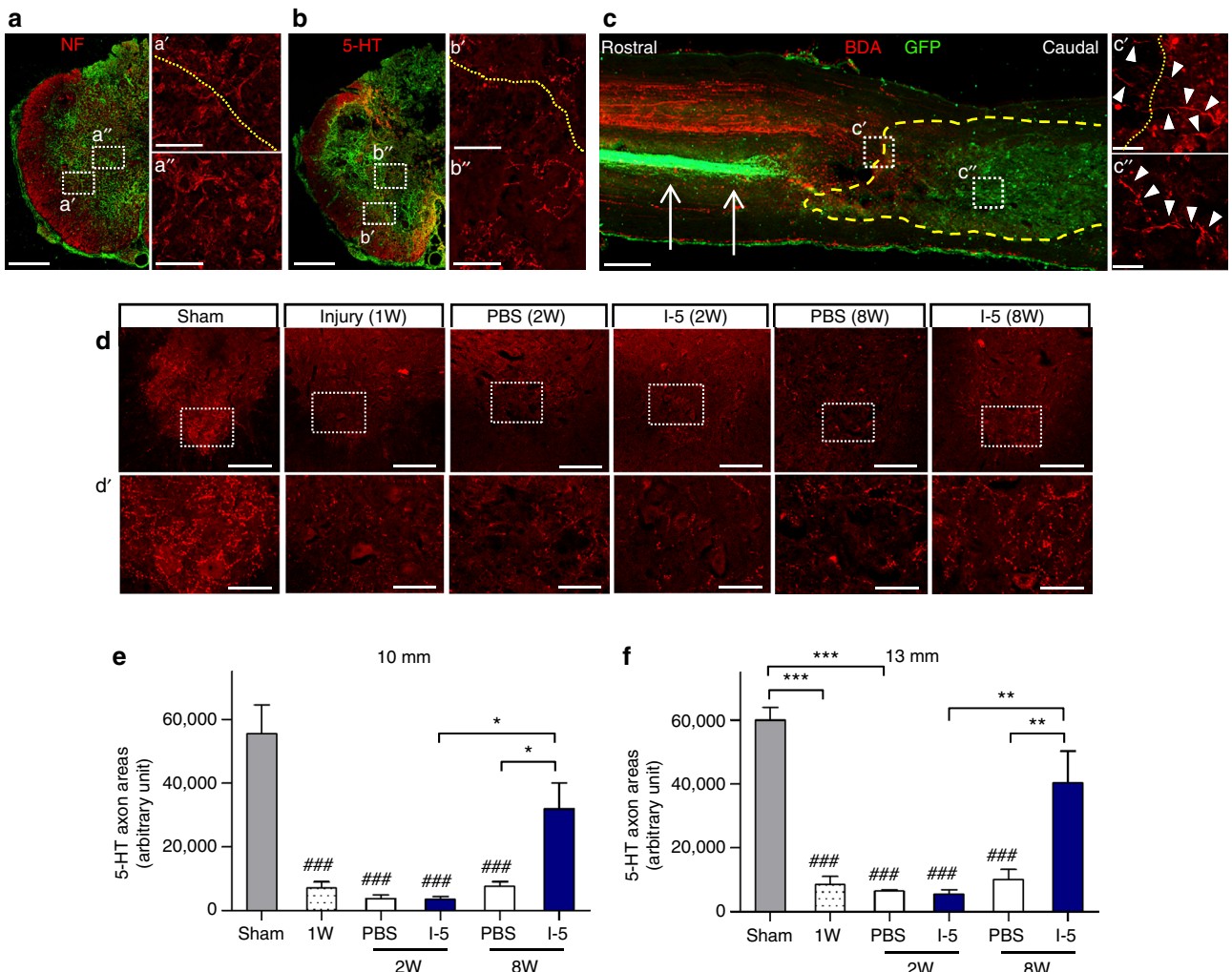

**Fig. 10** I-5 supports axon growth and promotes axonal reinnervation. **a**, **b** NF positive (**a**) and 5-HT (**b**) axon growth in the FN-rich matrix in the tansverse spinal cord sections at the epicenter of the lesion in animals with I-5 injection. *Red* indicates NF (**a**) or 5-HT (**b**) and *green* indicates FN immunoreactivity. The *boxed regions* in **a**, **b** are magnified in a′, b′ and a″, b″. The *dotted lines* in a′, b′ indicate the margins of the FN-rich matrix. *Scale bars* in **a**, **b** represent 500 µm. *Scale bars* in the magnified images represent 100 µm. **c** A representative image of the longitudinal spinal cord section from an animal with I-5 hydrogel. Descending axons were traced with AAV8-GFP and biotinylated dextran amine (*BDA*). The tissue section was stained with Alexa Fluor 594-conjugated streptavidin (*red*) and FN antibody (*green*). The corticospinal tract (*arrows*) was visualized by GFP signal. The *yellow dashed line* indicates the boundary of the FN-rich matrix. Boxed regions were magnified in c′, c″. Arrowheads indicate BDA-traced axons. The *scale bar* in **c** represents 500 µm and the *scale bars* in c′, c″ represent 50 µm. **d** 5-HT axon innervation in the ventral motor regions of the lumbar spinal cord in sham-operated animals, animals with injury alone sacrificed 1 week (W) after injury, animals with PBS or I-5 injection sacrificed 2W or 8W after injury. The boxed regions in **d** are magnified in d′. *Scale bars* in **d**, d′ represent 500 and 100 µm, respectively. **e**, **f** Quantification of 5-HT axon density in the ventral motor regions in the lumbar spinal cord. Quantification was done at 10 mm (**e**) or 13 mm (**f**) caudal to the epicenter. * and *** indicate $p < 0.05$ and $p < 0.001$ by one-way ANOVA followed by Tukey's post hoc analysis. ### indicates $p < 0.001$ compared to sham-operated group by one-way ANOVA followed by Tukey's post hoc analysis. $N = 5, 4, 4, 5, 9$ and 8 for the sham-operated, PBS (1W), PBS (2W), I-5 (2W), PBS (8W), or I-5 (8W) groups, respectively. *Error bars* represent the SEM

(CDCl3), δ (ppm): 0.8–1.0 (s, 6H), 1.1–1.3 (b, 3H), 1.3–1.6 (b, 2H), 1.6–1.9 (b, 1H), 2.5–2.8 (b, 4H), 2.6–2.9 (b, 49H), 2.8–3.3 (b, 2H), 3.4–3.8 (b, 75H), 3.9 (s, 1H), 4.0–4.3 (b, 5H), 6.9–7.2 (b, 1H), 7.4–7.6 (b, 1H), 7.9–8.1 (s, 1H).

**Preparation of I-5 hydrogel solution.** Ten percent (by weight) of the I-5 polymer in PBS solution was used for experiments after filtration using 0.2 μm cellulose acetate syringe filter. In the experiments where I-5 polymer solution had to be mixed with Taxol or siRNA nanoparticles, 20% (by weight) of I-5 solution was prepared.

**Measurement of the physical properties.** The structure of prepared polymers was estimated by a [1]H-NMR spectrometer (Varian Gemini-300, Agilent Technologies) operating at 400 MHz in the Fourier transform mode with CDCl₃ used as the solvent and by an FT-IR spectrometer (Spectrum GX FT-IR, Perkin-Elmer). The substituted amount of imidazole groups in IP was also determined by BCA

assay (Pierce). High performance of liquid chromatography (HPLC) system (Agilent, 1050 series) was also used to confirm the conjugation of imidazole. The molecular weight of I-5 was calculated by using a gel permeation chromatography system (Waters 1515, Waters) with a refractive index detector (Waters 2410) and two styragel columns (Waters styragel HR 4E and HR 5E) connected in line at a flow rate of 1 ml/min at 35 °C. THF containing 0.1% (by weight) tetra-butylammonium bromide was used as a mobile phase. Polystyrenes (MW: 1270; 3760; 12,900; 28,400; 64,200; 183,000; 658,000; 1,050,000; 2,510,000; 3,790,000) were used as standards. The viscosity of the aqueous I-5 polymer solution was measured by using a Brookfield RVDV-III + viscometer (Brookfield) between 5 and 70 °C under a fixed shear rate of 0.1 per s. The viscosity measurements were done with a set spindle speed of 0.2 rpm and with a heating rate of 0.33 °C/min. For in vitro stability test, 10% polymer solutions were mixed with a fluorescence dye Rhodamine B (at a concentration of 10 μg of 100 μ1), and 200 μ1 of labeled polymer solution was loaded onto a millicell (pore size 12.0 μm). The millicells

containing polymer solutions were incubated for 30 min in a heated oven at 37 °C. Then the millicells were moved to glass tubes containing 3 ml of PBS, and the tubes were incubated within a shaking incubator of which temperature is fixed at 37 °C.

**Evaluation of biocompatibility.** For evaluation of in vitro cytotoxicity, NIH3T3 cells were seeded in 96-well tissue culture plates at a density of $1 \times 10^4$ cells per well. After incubation for 24 h, the cells were washed with PBS. Then, 0.2 ml of the polymer solution was added to each well and the plates were incubated for 24 h at 37 °C in a humidified atmosphere with 5% $CO_2$. Cell viability was measured using the EZ-CYTOX assay kit (Dogen Bio) following the manufacturer's instructions. The absorbance was determined at a 450-nm wavelength using a microplate reader (Molecular Devices). For in vivo biocompatibility testing, 100 µl of 10% (by weight) was injected into the dorsal subcutis of BALB/c mice (6 weeks old, female). Body weight was monitored for 28 days and the existence of inflammatory signs in the tissue was examined at predetermined time points.

**Animals and surgical procedures.** All animal protocols were approved by the Institutional Animal Care and Use Committee of Ajou University School of Medicine. Adult female Sprague Dawley rats (250–300 g) were used in this study. Spinal contusion injury was inflicted using the Infinite Horizon impactor (Precision Systems and Instrumentation). Animals were anesthetized with 4% chloral hydrate (10 ml/kg, injected intraperitoneally), followed by a dorsal laminectomy at the 10th thoracic vertebral level (T10-11) to expose the dorsal surface of the spinal cord. Then, standardized contusion at a force of 200 kdyn was performed with the Infinite Horizon impactor. After the injury, muscles were sutured in layer and the skin was stapled. Bladder care was provided twice daily until spontaneous voiding resumed. I-5 injection was performed 1 week after contusion injury to avoid potential tissue damage related to the injection itself. I-5 hydrogel solution (10%) was maintained on ice until loaded into a 26 G Hamilton syringe (model 701RN) to prevent gelation. Animals were anesthetized and the dorsal surface of the spinal cord with the previous contusion injury was re-exposed. After the contused area was identified under bright illumination, the Hamilton syringe was advanced into the center of the contused area and 10 µl of the I-5 solution was injected manually. After injection, the syringe was kept in situ for 1 min to prevent regurgitation of the injected hydrogel through the injection site. When I-5 was injected with Taxol (1 µg/µl) or siRNA nanoparticles (see the section below), equal volumes of the agent and a 20% (by weight) solution of I-5 were mixed yielding a final gel concentration of 10% (by weight). For injection of AAV8-GFP for anterograde tracing of corticospinal axons, rats were placed on a stereotaxic frame, and a midline incision was made over the skull to disclose bregma. The skull overlying the right sensorimotor cortex was removed using a microdrill, and the AAV virus (AAV8 serotype with UbC promoter expressing GFP) was injected into the cortex through a 10 µl Hamilton microsyringe tipped with a pulled glass micropipette attached to a nanoliter injector. Ten injections (100 nl per site) were performed at a rate of 50 nl/min at the following coordinates: 1.5 and 2.0 mm lateral; 0.7 mm deep; 1.0 and 1.5 mm anterior; 0.3, 0.8, and 1.3 mm posterior to bregma. For BDA injection, a midline incision was made over the thoracic vertebrae and a laminectomy was performed to expose the underlying T6–T7 spinal cord and 10% BDA (10,000 MW; D1956, Invitrogen) tracer was injected into the right ventral gray matter (500 nl per site) using the following coordinates; 0.6 mm lateral to midline; depth, 1.5 and 2 mm. The needle was left in place for 1 min before moving to the next site. The brain and spinal cord were kept moist during the procedure and the skin was closed with sutures. Animals were sacrificed three weeks after external tracer injections.

**Knockdown of MMP-9 by siRNA nanoparticles.** To knockdown MMP-9 in in vivo spinal cord tissue, we used lipid nanoparticle-mediated siRNA delivery technology (Precision Nanosystems). Specifically, fluorescently labeled SUB9KITS nanoparticles were conjugated with candidate siRNA molecules with sequences targeting MMP-9. Validation of siRNA-mediated knockdown was performed using rat peritoneal macrophages. A total of $1 \times 10^5$ peritoneal macrophages were plated per well in a 24-well plate. After incubation for 24 h, the medium was removed and the cells were cultured with 1 µg/ml siRNA nanoparticles for 2 days. Total RNA was extracted using a PureLink RNA Mini Kit (Ambion) according to the manufacturer's instructions. cDNA was synthesized from 500 ng of total RNA using oligo dT primers and a PrimeScript 1st strand cDNA Synthesis Kit (Takara). Real-time quantitative PCR was conducted using SYBR Premix Ex Taq in an ABI 7500 system (Applied Biosystems). MMP-9 (NCBI: NM_031055) primers were designed as follows: forward: 5′-GGCCTATTTCTGCCATGACAAATAC-3′ and reverse: 5′-CTGCACCGCTGAAGCAAAAG-3′ (expected product size: 141 bp). The primers for ribosomal RNA used as a loading control were as follows: forward: 5′-CGCGGTTCTATTTTGTTGGT-3′ and reverse: 5′-AGTCGGCATCGTTTA TGGTC-3′ (expected product size: 240 bp). MMP-9 knockdown was also validated using immunocytochemistry. After 2 days of incubation with 2 µg/ml siRNA nanoparticles, peritoneal macrophages were fixed and the cells were stained with rabbit MMP-9 antibody (1:100; Millipore). For in vivo delivery, siRNA was reencapsulated with lipid nanoparticles to produce highly concentrated siRNA nanoparticles. For injection, 5 µl of 5 mg/ml nanoparticles were mixed with 5 µl of 20% I-5 hydrogel.

**Tissue processing and histochemistry.** For histological assessment of the lesion site, animals were sacrificed at 1 or 4 weeks after injection (2 or 5 weeks after injury) as early and late time points, respectively. Animals were anesthetized deeply with chloral hydrate and perfused intracardially with PBS, followed by 4% paraformaldehyde in 0.1 M phosphate buffer at pH 7.4. Then, the spinal cord was dissected and tissue blocks were postfixed in 4% paraformaldehyde for 2 h and then cryoprotected in a graded series of sucrose solutions. Twenty-µm-thick sections of the spinal cord were cut transversely using a cryostat (CM 1900; Leica) and thaw-mounted onto Super Frost Plus slides (Fisher Scientific). For morphological assessment of lesion cavities, serial spinal cord sections were subjected to eosin and eriochrome staining. The cross sections of spinal cord were immersed for 8 min in a staining solution consisting of 240 ml of 0.2% eriochrome cyanine RC (Sigma) and 10 ml of 10% $FeCl_3 \cdot 6H_2O$ (Sigma) in 3% HCl. The sections were then washed with running tap water, followed by 1% $NH_4OH$ for differentiation. After eriochrome cyanine staining, the sections were counterstained with eosin solution for better visualization of lesion cavities. For immunohistochemistry, spinal cord tissue sections were incubated overnight at 4 °C with primary antibodies: anti-GFAP (1:500; Dako, #0334), anti-fibronectin (1:100; Sigma, #F36480), Iba-1 (1:500; Wako, #019-19741), anti-PDGFR-β (1:300; Abcam), anti-collagen 1α1 (1:100; Santa Cruz, #sc-8784), anti-5-HT (5-hydroxytryptamine; 1:5000; Immunostar, #20080), anti-MMP-9, (1:100; Millipore, #AB19016), anti-CD45 (1:500; Bio-Rad, #MCA589R), anti-MBP (1:200; Abcam, #ab7399), CD206 (1:500; Abcam, #ab64693) or anti-CD11b (1:500; Bio-Rad, #MCA275R). After washing three times, the slides were incubated with appropriate secondary antibodies conjugated to the Alexa Fluor fluorescent dyes. For BDA staining, Alexa Fluor 594-conjuaged Streptavidin (1:500; Molecular Probes, #S32356) was used. Images were captured using a confocal laser-scanning microscope (Olympus). For brightfield imaging of GFAP immunoreactivities, spinal cord sections were incubated with anti-GFAP (1:2000; Dako, #0334) followed by biotinylated goat anti-rabbit IgG secondary antibody (1:400), and the antigen–antibody reaction was visualized using a Vectastain Elite ABC kit (Vector) with a Vector SG peroxidase substrate kit (Vector).

Tissue clearing and LFSM imaging of whole spinal cord tissue were performed as described previously[32]. Briefly, meninges of injured spinal cord were carefully removed and 8 mm-length of the spinal cord tissue centered on the epicenter region was cut. The samples were dehydrated in 50%, 80% and 100% THF solution on an agitating incubator at RT each for 3 h and lastly switched to fresh 100% THF for overnight. Next, THF solutions were switched to BABB (benzyl alcohol and benzyl benzoate) and the spinal cord tissue sample was continuously incubated on an agitating incubator until being transparent. After tissue clearing, longitudinal images were taken by LSFM (Ultramicroscope, Lavision Biotec) at a ×12.5 magnification with a 3 µm inter-image thickness. Based on autofluorescence signals from the spinal cord tissue, it was possible to delineate cystic spaces devoid of the tissue from the residual spinal cord. The boundaries of cystic cavities were manually drawn in every third image and the cystic cavities were highlighted in magenta color. Stacked images were made into a 3D movie using Imaris software (Bitplane).

**Quantitative image analysis and 3D reconstruction.** For quantitative analysis of cavity volume, serial spinal cord sections stained with eriochrome and eosin were three-dimensionally reconstructed. Three-dimensional reconstruction of the lesion cavity was done using the Neurolucida tracing software equipped with the 3D Slide Scanning Module (MBF bioscience). A total of 24 serial transverse spinal cord sections equally spaced 400 µm apart were used to create a 3D image corresponding to 1 cm-long spinal cord segment. Contours of the spinal cord outer boundary, the white matter, the cystic cavity, the intact gray matter, and the pathological spinal cord tissue (the area in which normal tissue architecture was not maintained) were manually drawn on each section, and then the Virtual Tissue software program generated 3D images. Different colors were assigned to distinguish the white matter (*light gray*), the gray matter (*green*), the pathologic spinal cord tissue (*yellow*), and the cystic cavity (*red*). The Neurolucida software calculated the volumes of the cystic cavities automatically. For quantification of Iba1-immunoreactive signal intensity, three regions of interest (ROIs) of identical size were placed at the dorsal, lateral, and ventral regions containing the borders between the residual white matter and damaged tissue with or without cavities at the epicenter. The lateral border of a dorsal ROI was placed immediately medial to the dorsal horn so that the dorsal ROI was located on the dorsal column. A lateral ROI was located just above the line crossing the intermediolateral horn. If the intermediolateral horn could not be identified due to the lesion, the lower border of a lateral ROI was placed just above the transverse midline of the spinal cord section. A ventral ROI was located below the ventral horn. If the ventral horn could not be located due to the lesion, the medial border of a ventral horn was placed 500 µm apart from the vertical midline of the spinal cord section. Then, the Iba1-immunoreactive signal above the predetermined threshold value was quantified using ImageJ software (publically available at http://imagej.nih.gov/ij/). For quantification of the number of motor neurons below the lesion site, pairs of transverse spinal cord sections (with an intersection interval of 40 µm) were selected 1.2, 1.6 and 2.0 mm caudal to the epicenter. The sections were stained with eriochrome and cresyl violet, and the surviving motor neurons with a longest cell diameter of at least 20 µm were identified and counted using a bright-field microscope (Olympus BX51). The average number of motor neurons in each pair was calculated. For quantification of MBP immunoreactivity in the residual white

matter, transverse sections 2 mm rostral to the epicenter, at the epicenter, and 2 mm caudal to the epicenter were selected and stained with anti-MBP antibody. Three ROIs of identical size were placed in the dorsal, lateral, and ventral white matter regions. The ROIs were located using the same criteria as the above ones for quantification of Iba1-immunoreactive signal intensity. The only difference was that they were placed within the MBP-positive residual white matter. Then, the MBP-immunoreactive signal above the predetermined threshold value was quantified using ImageJ software. Quantification of the 5-HT axon density in the caudal lumbar motor regions was performed as described previously[9] with a slight modification. Briefly, two transverse sections located 10 and 13 mm caudal to the epicenter were obtained from each animal. The number of pixels in the ventral motor regions occupied by the 5-HT fibers was quantified using ImageJ, and this value was normalized to the number of 5-HT fiber pixels in ventral motor regions rostral to the epicenter.

**Zymography**. For detection of MMP-2 and MMP-9 enzymatic activities, PBS ($N = 5$) and I-5 ($N = 4$) injected animals were sacrificed 1 week after injection (2 weeks after injury). The 1 cm-long spinal cord segment with the epicenter at its middle was freshly dissected and quickly frozen at $-70\,^{\circ}\text{C}$. The spinal cord tissue was homogenized and sonicated in RIPA buffer. Fifty µg of the protein sample was loaded onto a polyacrylamide gel containing SDS and gelatin and subjected to electrophoresis. The gel was re-natured in renaturing buffer (2.5% Triton X-100) to allow proteins to regain their enzymatic activities, and then washed three times with developing buffer containing 50 mM Tris-HCl, 200 mM NaCl, 5 µM $ZnCl_2$, 5 mM $CaCl_2$, and 0.2% Brij-35 (Sigma). The gel was transferred to fresh developing buffer and incubated at $37\,^{\circ}\text{C}$ for 72 h. Then, the gel was stained with Coomassie blue for 2 h followed by destaining in methanol and formic acid. The intensity of clear bands resulting from protease digestion was determined by densitometry using ImageJ software.

**In vitro assay for interaction between macrophages and I-5**. Nile Red (Sigma-Aldrich), a hydrophobic fluorescent dye, easily interacts with the hydrophobic core part of polymeric micelles via a hydrophobic interaction and induces a size increase of the nano-scaled micelles[58]. When 80 µg of polymer solution was mixed with 0.1 µg Nile Red and sonicated for 1 h, the mixture formed particulate micelle structures. The formation of nanoparticles made of polymer micelle structures was confirmed by a size increase after addition of Nile Red from 30 to 34 nm, as measured using a Zetananosizer (Zetasizer Nano ZS, Malvern Instruments Ltd.). RAW 264.7 cells (purchased from ATCC; ATCC TIB-71) were seeded in 24-well plates ($2.5 \times 10^4$ cells per well) and incubated overnight in complete DMEM culture medium. Then, the culture medium was replaced with 1 ml of serum free medium containing the nanoparticles composed of Nile Red and polymer with or without the imidazole group. After 1 h, 1 ml of complete medium was added to every well and the cells were incubated for 2 h. After washing with PBS, fluorescence images of the cultured macrophages were captured using a confocal microscope (Olympus). The intensity of Nile Red fluorescence was quantified using a fluorescence spectrophotometer (Synergy H1 Multi-mode). To determine if the interaction was mediated by histamine receptors, mepyramine maleate (Santa Cruz), an inhibitor of histamine receptor 1 (H1), or JNJ7777120 (Santa Cruz), an inhibitor for histamine receptor 4 (H4), were added at a concentration of 20 µM to cultured macrophage cells 30 min before adding the nanoparticles.

**Behavioral assessment**. The number of animals required for behavioral analyses was determined based on the original report of the BBB open field locomotor test where 5–9 animals per group were used[49]. Animals that showed the BBB score 3 and higher at 1 day after contusion injury were excluded. One subject in PBS and two subjects in I-5 injection group were excluded. Immediately after contusion injury, animals were randomly allocated to either PBS ($N = 9$) or hydrogel ($N = 8$) injection group. To ensure blind assessment of behavioral recovery, animals were assigned new identification codes after the injection 1 week after injury by an independent experimenter who was not involved in either animal surgery or behavioral assessment. The original identification codes had been available only to the independent experimenter until all the behavioral experiments and assessments (including catwalk analysis) were completed. Locomotor recovery was evaluated using the BBB open field locomotor scale and Catwalk footprint analysis (Noldus Information Technology). Rats were allowed to walk freely in an open field and the locomotor rating scale was determined after a 3-min observation session. Recovery of hindlimb movements was assessed 1 day after injury, 7 day after injury right before injection, and then once a week for a duration of 8 weeks. For Catwalk gait analysis, animals were first trained to walk on the Catwalk runway in an uninterrupted manner. On the test day, four runs without significant interruptions per animal were obtained as valid runs. Individual footprints were determined manually using Catwalk software. Then, the software automatically calculated five gait parameters. The angle of hindpaw rotation was defined as the angle (in degrees) of the hindpaw axis relative to the runway axis. An increase in rotation angle indicates external rotation of the hinpaws. The base of support was measured as the width of the area between the left and right hindpaws. Values from both hindpaws were averaged to calculate the stride length and paw angle values. The footprints of hindpaws tend to overlap those of forepaws during walking in uninjured animals.

However, injured animals often lose this coordination between the hind- and forepaws[59]. Therefore, the relative position of the fore- and hindpaws was obtained by directly measuring the distance between the center pads of ipsilateral fore- and hindpaws in each step cycle. Regularity index is used for an objective analysis of gait coordination and calculated from the number of normal step sequence patterns multiplied by four and divided by the total amount of paw placements[43].

**Statistical analysis**. Statistical analysis was performed using GraphPad Prism software (version 5.0). Unpaired Student's $t$-test (two-tailed) was used to compare mean values of two groups. One-way ANOVA followed by Tukey's *post hoc* analysis was used for the mean comparison of three groups and more. Repeated-measures two-way ANOVA was used to compare differences in the BBB scores matched at different time points.

**Data availability**. All data is available from the authors upon reasonable request.

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

## Acknowledgements

This research was supported by Korea Institute of Science and Technology (2E26230, S.-C.S.), National Research Foundation of Korea (2014M3A9B6034220, S.-C.S.; 2014M3A9 B6034224, 2015R1A2A1A01003410, B.G.K.), and NINDS R01NS081040 (J.K.L.).

## Author contributions

L.T.A.H., Y.-M.K., D.H.H., J.K.L., S.-C.S. and B.G.K. planned research and designed experiments. L.T.A.H., Y.-M.K. and H.H.P. performed and led the majority of experiments. H.H.P., D.H.H., Y.C., E.M.L., S.Y. and J.K.L. participated in data acquisition. L.T.A.H., Y.-M.K., H.H.P., J.K.L., S.-C.S. and B.G.K. interpreted data. L.T.A.H., Y.-M.K., H.H.P., S.-C.S. and B.G.K. wrote manuscript. S.-C.S. and B.G.K. approved manuscript submission.

## Additional information

**Competing interests:** The authors declare no competing financial interests.

