## [Peer Review file · Nature Communications]

Reviewers' comments:

Reviewer #1 (Remarks to the Author):

Overview

The topic of the communication submitted is an important development in the treatment of spinal cord injury, and is certainly relevant to Nature Communications. Not only the reduction in cyst cavities, but the formation of fibronectin ECM within the cyst is a major development for SCI that can be used as a basis for future regenerative therapies. If accepted, this would be a major news story in my opinion. Therefore, it is very important – especially for the authors involved – that this approach is robust and reproducible. I would also highly recommend that the senior authors on this manuscript look carefully at all the IHC data and slides themselves, as once published this will attract significant interest and will be reproduced in laboratories across the world. It is imperative to the authors that all aspects of this study are watertight. However, the polymer synthesis description, schematic and characterization are inadequate and are a major barrier to publication.

Comment 1

Seven days after the initial SCI, there will already be a cyst formed in the spinal cord. I have personally made similar injuries to the spinal cord of Wistar rats (250-300g), and very distinct, large cysts exist six days after initial injury (not just “discernable” as mentioned in line 126 but very distinct by IHC using GFAP/CD68+). It would be an excellent addition to this manuscript to show the extent of cyst formation prior to injection, rather than something from literature stating what it should be. This will help the reader understand into what exactly what environment the hydrogel is injected, and then how this environment is altered by the hydrogel.

Comment 2

The mechanisms of action proposed by the authors have been seriously considered and thought through, with many controls and aspects investigated. However there is one aspect that I find difficult to understand.

I can fully appreciate how injecting this polymer into the cyst could affect the macrophages within. What I cannot understand is how GFAP is later seen within the cysts...where did the astrocytes come from? Did they migrate? From my own observations on personally performed contusion injuries, this cyst is almost exclusively filled with CD68+ macrophages, and NO GFAP labeling, after six days. Here the injection was done at 7 days, and although Sprague Dawley are used, there cannot be such a difference in the two species.

So I doubt that GFAP-positive astrocytes could exist within the cyst, and I really question the GFAP staining that was performed in conjunction with the FN staining.

I noticed that the GFAP anti-body used was the one from DAKO – this is one of the most robust antibodies that I have used, and the level of dilution (1:500) was not ideal in this study, perhaps increased for use of fluorescent 2ndary antibodies. Previously I used 1:2000 or even 1:5000 dilutions, which were ideal for staining the GFAP whilst providing no background staining.

Furthermore, I used a permanent chromophore (3,3'-diaminobenzidine tetrahydrochloride). The images are excellent and clear as to the distribution of GFAP within the spinal cord.

Since the authors are obviously effective at performing IHC, I would like you to use a section of the I-5 injected cords, using this lower dilution of GFAP and a permanent chromophore (with negative controls – no antibody), to ascertain whether the GFAP staining in Figure 4 D&J that you have is real or an artifact (GFAP could also have some non-specific binding to the injectable hydrogel). I also suggest to use the permanent chromophore, so that you have proof that this is indeed the case (it does not fade is very good to bring out at a later date as proof, unlike fluorophores that fade with time. I find it very strange to have the GFAP present in the injury epicenter, after a cyst has formed (it was 1 week after injury when the injections were made – the cyst has already formed). For me this is the biggest issue with the manuscript that I can't reconcile with logic. Fortunately for the authors, this is a simple and quick experiment to perform, that increases the robustness of the manuscript.

Comment 3

Acronyms for words have been defined multiple times throughout the manuscript: Please correct – these should be defined once and then used in their abbreviated form. Repeated definitions of

acronyms include:

BBB, MBP, ROIs, FN, SEM, MMP-9, I-5 ...there are probably more repeated definitions of acronyms.

GFAP was not defined, NF was defined at the end, but only used once and full word previously used once. IH was defined, then used only once, probably unnecessarily – no need to abbreviate these words.

Comment 4

There is a strange jump (single data point) in the rheology of the I-5 and also the CP-2 in the supplementary. It appears that this rheology data has been performed only once, and these artifacts remain in the rheology. This is a simple task to perform – please repeat multiple times.

Comment 5

The authors do not appear to distinguish the difference between scaffolds and matrices. These are two terms to describe different biomaterial classes – these have been previously defined by David Williams, who is the gold standard with biomaterial definitions. Here, and also self-assembling peptides – are injectible matrices, not scaffolds. Please correct the reference to the self-assembling peptides.

Comment 6

There is a reference in line 321 that portrays other injectible, self-assembling, peptide matrices as only being “mere physical scaffolding”. Actually, from my conversations with these authors of previous studies also purport there is a neuroregenerative aspect to the matrices. Furthermore, I believe the data that the authors provide is compelling enough to stand alone, and that it is not necessary to diminish the quality of other studies using injectible matrices to raise the quality of yours. Actually, I believe that there is also a physical support aspect to the in-situ gelling hydrogel used in this study. Otherwise, modifying a soluble polymer with these moieties and injecting it would also be effective.

Comment 7

There have also been cell-invasive scaffolds implanted into the spinal cord (also in the non-clinically relevant hemi-section models), that also notice a reduction in cyst formation. While these studies are not as impressive as the one performed here, and are likely due to physical stabilization after injury, they do have some relevance with respect to reduced cyst formation.

Some are very old – Woerly, Plant, but also:

A. Bakshi et al., Mechanically engineered hydrogel scaffolds for axonal growth and angiogenesis after transplantation in spinal cord injury. *J Neurosurg-Spine* 1, 322-329 (2004).

V. R. King et al., Characterization of non-neuronal elements within fibronectin mats implanted into the damaged adult rat spinal cord. *Biomaterials* 27, 485-496 (2006).

A. Jain et al., In situ gelling hydrogels for conformal repair of spinal cord defects, and local delivery of BDNF after spinal cord injury. *Biomaterials* 27, 497-504 (2006).

H-Y Li et al. Host reaction to poly(2-hydroxyethyl methacrylate) scaffolds in a small spinal cord injury model. *J Mat Sci Mater Med*, 24, 2001-2011. (2013)

Please look at mentioning these papers and how cysts have been reduced after SCI when implanted in non-clinically relevant models. Note that this does not diminish this study, since here the intervention is performed on contusion injuries and in a clinically relevant manner.

Comment 8

Line 338-339 – the sentence “Thus, it is conceivable that I-5 injection promotes ECM remodeling in the rat spinal cord in a way that emulates the characteristics of the ECM in the mouse spinal cord.” Is both unclear and overly speculative.

Comment 9

It is important for many reasons that the entire study can be reproduced by others. While many of the IHC work is standard and can be performed readily, the most challenging issue with respect to reproduction is the synthesis of the polymer. I am not convinced that all the information provided with this respect has been provided. No value for the polymer concentration was provided. How was it sterilized – filtration? The reaction scheme in Figure 1 could be much clearer, and the HNMR. It appears that while some information is provided, not enough is provided to truly let some other group reproduce this research.

Comment 10

My final comment is that the authorship team must provide better description of synthesis and characterization of the polymer. Polyphosphazenes are notoriously difficult to control in synthesis and the information required for other to repeat the synthesis of the polymer is not accurate nor sufficient. As it stands, this work cannot be reproduced, since it lacks sufficient information on even what the material is – the 18771 Da molecular weight (Mw is never presented as accurately as this – 18.7 kDa or approx. 18 kDa is better due to natural errors in the analysis) shows that this was a n=1 synthesis, so reproducibility is a major problem here. What were the results of multiple batches? Furthermore, hydrophilic polyphosphazenes hydrolyze quite quickly, and therefore some stability data is required.

This lack of description/standardization of the hydrogel is such a critical issue, that if not fixed, would lead to me recommend that this study is rejected. It is not satisfactory, that a irreproducible polymer with properties that are the primarily reason for the results, is described in this manner. If the research community are unable to accurately synthesize the polymer, then the authors have amazing results that are unrepeatably to anyone else but themselves. I would ask for a more accurate synthesis description that achieves the minimum standards for publication in even low-impact factor journals. Currently the description is not sufficient for publication in ANY journal.

Comment 11

Except for the polymer synthesis and rheological characterization using n=1, the number of experiments and their description is statistically satisfactory

Reviewer #2 (Remarks to the Author):

The paper is a throughout investigation of the effects caused by multiple injections of a synthetic hydrogel into a contusive SCI injury model.

The work is well described however I would suggest some additional experiments to strengthen its impact.

Page 5 synthesis section: the authors mention the results of a pilot experiment. It would be helpful for the reader to look at such not published data because comparison with other not satisfactory hydrogels is informative. I suggest to place these data in the SI.

Is the hydrogel bioabsorbable? Is it still present at 4 weeks after treatment?

Did the author run any previous mass loss test in vitro? If so, a measurement of the bioabsorption time would be interesting. Is the bioabsorption influenced by the chosen functionalization? I think so but authors should better explain that.

Also they may add a few lines about the possible mechanisms of degradation in vivo.

In order to erase any doubts about collateral sprouting of the spared nervous tracts, instead of regenerating fibers crossing the lesion, neural tracing assessment is recommended. A possible scenario is intracranial injections (over sensorimotor cortex) of anterograde tracers and assessment of axons labeling after 2 weeks at the site of implant.

The description of the criteria for the selection of the ROI seems to be quite discretionary but it is a critical parameter that may influence results. Please clarify this point.

Figure 2a: what is the kinetic of the gelation process? A viscosity vs time graph at fixed 37 C would be appreciated instead of a qualitative set of pictures. Also, the movie is not clear as an opaque material seems to appear in the +4C setting as well.

In figure 2b the rod-like material cannot be seen. Please provide an enlarged image or mark the material with a dye so as its shape can be better appreciated.

Figure 8: please add the BBB score of the sham group. Also the authors should better clarify if the treatment was given at day 0 or at day 7 of the timescale. Lastly, please indicate in the graph the

BBB value of the animals after injury and right before treatment.

Reviewer #3 (Remarks to the Author):

Review Nature Communications: NCOMMS-16-26305

An injectable hydrogel enhances tissue repair after spinal cord injury by promoting extracellular matrix remodeling

Reviewer's summary:

The manuscript describes the use of an injectable hydrogel to reduce cystic cavitation following spinal cord injury. The authors shortly describe the synthesis and main characteristics of the hydrogel prior to investigating its effect on cavity formation in a contusion type injury. They observed a replacement of the fluid filled cavity with fibronectin rich, GFAP-negative scar tissue and an increase in spared tissue. The remodeling of the injury site was partly mediated by MMP9, which was upregulated following hydrogel injection. Blocking the effect of MMP9 with siRNA lead to an increase in cavity size. The authors further demonstrate that the positive effects were mediated by an imidazole moiety, as hydrogels lacking this group demonstrated less positive effects.

Reviewer's comments:

I enjoyed reading the manuscript and the development of injectable hydrogels to promote tissue repair in the central nervous system is an interesting topic and well suited for the readers of nature communications. The authors demonstrated nicely that MMP9 and the imidazole moiety are positively involved in scar formation and it is interesting that the fibrotic scar can lead to neuroprotection and functional improvements.

From my point of view, three main aspects could be further developed to strengthen the manuscript:

(1) The scar tissue formed after injury is an improvement over the fluid filled cavity, but the authors mention that it is mainly composed of fibroblasts and macrophages. Plotting the pathological tissue, white matter and gray matter additionally to the cavity size would be interesting to gain more insight on the tissue response following injury and treatment.

In addition, calculating the volume enclosed by host astrocytes could give further information on the ratio of neural vs scar tissue.

(2) M2, rather than M1, macrophages have been thought to be beneficial for tissue remodeling. The author could provide further insight on the observed remodeling process by analyzing if there was a shift towards M2 macrophages (e.g. CD206) at the lesion site following hydrogel injection.

(3) How did the authors conclude that the axons found in the caudal spinal cord were reinnervated rather than spared axons? The differentiation seems to be hard to make.

Furthermore, it would be interesting to demonstrate that axon cross the fibronectin rich area, this could be done using e.g. retrograde tracing.

Additional points include:

The swelling of the material is not mentioned, if the swelling was characterized previously, a simple reference would be sufficient.

How was decision made to inject 10µl?

Fibroblasts express fibronectin, as demonstrated, which can promote axonal outgrowth.

Additionally, they express many growth factors, which could have acted neuroprotective. This could be discussed.

Please plot the irregularity index as a measure of interlimb coordination. (see Koopmans, G.C., Honig, W.M.M., Hamers, F.P.T., Steinbusch, H.W.M., and Joosten, E.A.J. (2005), The assessment of locomotor behavior in spinal cord injured rats; the importance of objective analysis of locomotion. J. Neurotrauma 22, 214-225)

The amount of detail provided for the material and methods seems sufficient for reproduction.

Responses to the reviewers' comments

Reviewer #1

The topic of the communication submitted is an important development in the treatment of spinal cord injury, and is certainly relevant to Nature Communications. Not only the reduction in cyst cavities, but the formation of fibronectin ECM within the cyst is a major development for SCI that can be used as a basis for future regenerative therapies. If accepted, this would be a major news story in my opinion.

Comment 1) Therefore, it is very important – especially for the authors involved – that this approach is robust and reproducible.

Responses: To demonstrate reproducibility of the bridging effects by I-5, we sought for an independent evaluation in a laboratory different from the place where the I-5 injection was performed. Since one of our coauthors (JKL) has an expertise in a macroscopic imaging with tissue clearing technique, we shipped paraformaldehyde-fixed spinal cord tissues to his laboratory. Whole spinal cord tissues were cleared and imaged using the light sheet fluorescence microscope (LSFM) following the protocol published recently¹. The images were reconstructed as movies and inserted as **Movie S3 and S4**. We argue that the 3D imaging of whole spinal cord tissue at an independent laboratory provides a further support for the reproducibility of our findings.

Descriptions on the results of this experiment were inserted in the text (page 8, line 15). In addition, we added brief descriptions on this imaging method (page 31, line 11).

I would also highly recommend that the senior authors on this manuscript look carefully at all the IHC data and slides themselves, as once published this will attract significant

interest and will be reproduced in laboratories across the world.

It is imperative to the authors that all aspects of this study are watertight.

Comment 2) *However, the polymer synthesis description, schematic and characterization are inadequate and are a major barrier to publication.*

Responses: To address the reviewer's criticism, we tried our best to provide detailed information on the synthesis and characterization of I-5 hydrogel. We also added more detailed information in the synthetic schematics contained in Fig. 1. Point-to-point descriptions on several issues are found in the responses to the comments 11, 12, and 13) below (page 11 ~ page 15 in this letter).

Comment 3) *Seven days after the initial SCI, there will already be a cyst formed in the spinal cord. I have personally made similar injuries to the spinal cord of Wistar rats (250-300g), and very distinct, large cysts exists six days after initial injury (not just "discernable" as mentioned in line 126 but very distinct by IHC using GFAP/CD68+). It would be an excellent addition to this manuscript to show the extent of cyst formation prior to injection, rather than something from literature stating what it should be. This will help the reader understand into what exactly what environment the hydrogel is injected, and then how this environment is altered by the hydrogel.*

Responses: In response to the reviewer's comment, we newly generated 4 animals that were injured and sacrificed 7 days after, the time point when I-5 hydrogel is supposed to be injected. As shown in Fig. S3 in the revised manuscript, distinct cystic cavities were consistently observed at this time point. However, the size of cystic spaces were usually smaller than that observed at the 5-week time point after injury (4 weeks after PBS injection). In particular, rostrocaudal extent of cystic lesions was quite limited compared to that observed at the 5-week time point. The average cavity volume at the 1-week time

point was 0.46 mm³ (quantified using NeuroLucida software, datum not included in the manuscript), approximately 50% of the cavity volume at the 5-week time point. Previous studies also reported progression of cystic lesions between days' and weeks' time points after injury. A mild injury (12.5 mm drop) using NYU impactor in Long Evans rats resulted in ill-defined lesions with little cavity spaces at 8 days but the lesions progressed to form clear cystic cavities at 21 days². Similarly, Sprague-Dawley rats developed early stage of cystic lesion at 1 week and showed almost fully matured cystic spaces by 4 weeks after a contusion injury using a customized impactor device³.

We also performed immunofluorescence staining with both GFAP and CD68 antibodies and found that non-cystic, eosin-stained region found at the epicenter was filled with CD68 (ED-1) positive macrophages surrounded by GFAP positive astrocytes (Fig. S3 in the revised manuscript). Hill et al., also observed that the injury epicenter was filled with densely packed macrophages at 8 days after injury². It is conceivable that the lesion epicenter starts to form cystic spaces at several days after injury, but the cystic lesions may be largely filled with packed macrophages. These macrophage-filled areas may be replaced by fluid-filled, fully matured cysts by several weeks after injury. In this line of thoughts, it would be appropriate to consider that very distinct cysts, although filled with packed macrophages, can be formed 6 or 7 days after injury. We assumed that the large cysts the reviewer observed in his or her own study might represent the macrophage-packed cystic lesions similar to that shown in Fig. S3b).

In the revised manuscript, we added Fig. S3 to address the above issue and added several sentences in the text explaining the Fig. S3 (page 7, line 14).

***Comment 4)** The mechanisms of action proposed by the authors have been seriously considered and thought through, with many controls and aspects investigated. However there is one aspect that I find difficult to understand.*

I can fully appreciate how injecting this polymer into the cyst could affect the macrophages within. What I cannot understand is how GFAP is later seen within the

cysts... where did the astrocytes come from? Did they migrate? From my own observations on personally performed contusion injuries, this cyst is almost exclusively filled with CD68+ macrophages, and NO GFAP labeling, after six days. Here the injection was done at 7 days, and although Sprague Dawley are used, there cannot be such a difference in the two species.

So I doubt that GFAP-positive astrocytes could exist within the cyst, and I really question the GFAP staining that was performed in conjunction with the FN staining.

I noticed that the GFAP anti-body used was the one from DAKO – this is one of the most robust antibodies that I have used, and the level of dilution (1:500) was not ideal in this study, perhaps increased for use of fluorescent 2ndary antibodies. Previously I used 1:2000 or even 1:5000 dilutions, which were ideal for staining the GFAP whilst providing no background staining. Furthermore, I used a permanent chromophore (3,3'-diaminobenzidine tetrahydrochloride). The images are excellent and clear as to the distribution of GFAP within the spinal cord.

Since the authors are obviously effective at performing IHC, I would like you to use a section of the I-5 injected cords, using this lower dilution of GFAP and a permanent chromophore (with negative controls – no antibody), to ascertain whether the GFAP staining in Figure 4 D&J that you have is real or an artifact (GFAP could also have some non-specific binding to the injectable hydrogel). I also suggest to use the permanent chromophore, so that you have proof that this is indeed the case (it does not fade is very good to bring out at a later date as proof, unlike fluorophores that fade with time. I find it very strange to have the GFAP present in the injury epicenter, after a cyst has formed (it was 1 week after injury when the injections were made – the cyst has already formed). For me this is the biggest issue with the manuscript that I can't reconcile with logic. Fortunately for the authors, this is a simple and quick experiment to perform, that increases the robustness of the manuscript.

Responses: We thank the reviewer for carefully inspecting the authenticity of our immunostaining results. We reviewed all the immunostaining data with both GFAP and FN antibodies in the spinal cord sections obtained at 1 week and 4 week after injection (2 week and 5 week after injury). We found very little GFAP staining signal within FN

positive matrix in most of the sections we reviewed. In only a few slides we observed remarkable GFAP staining as intense as that presented in Fig. 4J-K in the original version of our manuscript, and it turned out to be just a background signal not specific to astrocytes. Therefore, we replaced the representative figures at the 4-week time point in Fig. 4 with the figures that do not exhibit nonspecific staining in the revised manuscript (Fig. 4j-k in the revised manuscript).

However, we did find occasional GFAP-positive astrocytes remaining within the FN-rich matrix. To examine whether the GFAP immunoreactivities shown by fluorescence signal represent genuine presence of astrocytes, we performed GFAP immunostaining using 1:2000 dilution of antibody from DAKO and DAB as a chromogen as the reviewer recommended. As are shown in the figures below, GFAP immunoreactivities, visualized by the method suggested by the reviewer, are recognized at 4 weeks within the area surrounded by dense GFAP-positive astrocytic scars. At 1 week after hydrogel injection, the boundary of GFAP-positive astrocytic scars was not as discrete as that at 4 weeks. Nonetheless, GFAP immunoreactivities stronger than those found in the boxed regions at 4 weeks were frequently observed in the central epicenter region at this time point. This is consistent with the findings we reported in fig. 4 using double immunofluorescence staining with GFAP and FN antibodies. Therefore, we conclude that there are a small number of GFAP-positive astrocytes within the FN-rich matrix at 4 weeks after injection. It is highly likely that a larger number of astrocytes reside in the central epicenter region at 1 week after injection and those astrocytes move out to the periphery of the epicenter as FN-rich matrix matures in the central epicenter between the 1- and 4-week time points. Therefore, the remaining astrocytes within FN-rich matrix may represent a population that have not migrated out or been repulsed by fibroblasts rather than those that have come or migrated from elsewhere in the injured spinal cord.

To provide information on GFAP-positive astrocytes within FN-rich matrix, we inserted in Fig. 3 the images of transverse spinal cord sections stained with GFAP according to the protocol suggested by the reviewer in the revised manuscript.

Comment 5) Acronyms for words have been defined multiple times throughout the manuscript: Please correct – these should be defined once and then used in their abbreviated form. Repeated definitions of acronyms include:

BBB, MBP, ROIs, FN, SEM, MMP-9, I-5 ...there are probably more repeated definitions of acronyms.

GFAP was not defined, NF was defined at the end, but only used once and full word previously used once. IH was defined, then used only once, probably unnecessarily – no need to abbreviate these words.

Responses: According to the reviewer’s recommendation, we provided definitions of the acronyms for BBB, MBP, ROIs, FN, SEM, MMP-9, and I-5 only once where the full name of each acronym appears first. The full name of GFAP, glial fibrillary acidic protein, was inserted. The full name of NF was inserted at the first part. The acronym for Infinite Horizon (IH) impactor was deleted and only the full name was given in the revised manuscript.

***Comment 6)** There is a strange jump (single data point) in the rheology of the I-5 and also the CP-2 in the supplementary. It appears that this rheology data has been performed only once, and these artifacts remain in the rheology. This is a simple task to perform – please repeat multiple times.*

Responses: In accordance with the reviewer’s suggestion, we repeated the viscosity measurement four times for both I-5 and CP-2. However, we still observed unexpected rises of the mean viscosity value at temperatures higher than 40°C, although there was a large variation between the 4 replicate measurements. There is no obvious answer at this moment as to why the strange jumps occur at a higher temperature range. We have occasionally found such unexplained hikes in other poly(organophosphazenes)-based hydrogels⁴. However, these unexpected viscosity increases occurred in all occasions at a very high temperature range that is not physiological. In addition, we did not experience any unexpected physical properties related to gelation in hydrogels showing the strange jumps in the viscosity curve.

We replaced the viscosity curve in figure 2 with the new one where the average values from 4 replicate experiments were obtained. We also change the viscosity curve in figure S6 for CP-2.

***Comment 7)** The authors do not appear to distinguish the difference between scaffolds and matrices. These are two terms to describe different biomaterial classes – these have been previously defined by David Williams, who is the gold standard with biomaterial definitions. Here, and also self-assembling peptides – are injectable matrices, not scaffolds. Please correct the reference to the self-assembling peptides.*

Responses: We are grateful for the reviewer’s advice on selection of terminology for biomaterials. We tried to incorporate the reviewer’s advice in as many parts of the manuscript as possible.

For example, in the second and third sentences of the third paragraph in the introduction (page 3, line 16), we described both scaffolds and matrices denoting previous approaches using biomaterials as shown below,

“Implanting various tissue-engineered **scaffolds or matrices** has been reported to reduce cyst formation. Since in most cases of human SCIs, the injuries are incomplete with a significant portion of the white matter spared, surgical procedures involving the implantation of **scaffolds or matrices** are prone to aggravating functional deficits.”

In the first sentence of the second paragraph in the discussion section (page 18, line 11), we designated injectable biomaterials used in previous studies as “**injectable matrices**” instead of injectable hydrogels to incorporate the reviewer’s insight on this issue. In the next sentence, we replaced the term “self-assembling peptide” with the “**self-assembling matrices**”.

Comment 8) There is a reference in line 321 that portrays other injectable, self-assembling, peptide matrices as only being “mere physical scaffolding”. Actually, from my conversations with these authors of previous studies also purport there is a neuroregenerative aspect to the matrices. Furthermore, I believe the data that the authors provide is compelling enough to stand alone, and that it is not necessary to diminish the quality of other studies using injectable matrices to raise the quality of yours. Actually, I believe that there is also a physical support aspect to the in-situ gelling hydrogel used in this study. Otherwise, modifying a soluble polymer with these moieties and injecting it would also be effective.

Responses: We did not intend to state that the effects of self-assembling matrices described in the discussion section are generated by “mere physical scaffolding” function. The point of referring to physical scaffolding ability was that the scaffolding function would be considered as a sort of primary or essential element of any biomaterial scaffolds

or matrices, and the striking bridging effects of I-5 would be explained by an extra function, that is its dynamic interaction with host cells and/or matrices, in addition to the primary scaffolding effect. Furthermore, we fully agree to the reviewer's comment that there is a physical supporting function that contributes to the effects of I-5. We already mentioned this aspect in the lines from 371 to 375 in the original manuscript (page 21, line 5) referring to a possibility that the formation of fibrotic matrix may increase structural stability of post-injury spinal cord.

We admit that the expression “mere physical scaffolding” may provide potential readers with an unintended impression that the scaffolding function is inferior or unimportant. Therefore, we revised the corresponding sentence as shown below (page 18, line 20),

“Therefore, the superior bridging effects of I-5 hydrogel likely result from its dynamic interaction with cellular components and/or interstitial matrix in the host tissue **in addition to its primary function of physical support or structural stabilization.**”.

Comment 9) There have also been cell-invasive scaffolds implanted into the spinal cord (also in the non-clinically relevant hemi-section models), that also notice a reduction in cyst formation. While these studies are not as impressive as the one performed here, and are likely due to physical stabilization after injury, they do have some relevance with respect to reduced cyst formation. Some are very old – Woerly, Plant, but also:

A. Bakshi et al., Mechanically engineered hydrogel scaffolds for axonal growth and angiogenesis after transplantation in spinal cord injury. J Neurosurg-Spine 1, 322-329 (2004).

V. R. King et al., Characterization of non-neuronal elements within fibronectin mats implanted into the damaged adult rat spinal cord. Biomaterials 27, 485-496 (2006).

A. Jain et al., In situ gelling hydrogels for conformal repair of spinal cord defects, and local delivery of BDNF after spinal cord injury. Biomaterials 27, 497-504 (2006).

H-Y Li et al. Host reaction to poly(2-hydroxyethyl methacrylate) scaffolds in a small

spinal cord injury model. J Mat Sci Mater Med, 24, 2001-2011. (2013)

Please look at mentioning these papers and how cysts have been reduced after SCI when implanted in non-clinically relevant models. Note that this does not diminish this study, since here the intervention is performed on contusion injuries and in a clinically relevant manner.

Response: According to the reviewer's recommendation, we added the references in the introduction section where we added one sentence shown below,

“Implanting various tissue-engineered scaffolds or matrices has been reported to reduce cyst formation.” Four references including the two the reviewer recommended (Bakshi et al.,; King et al.,) were cited at the end of the above sentence. The reference by Jain et al., was already cited at the beginning of the discussion section in the original manuscript.

In the reference by Li et al., reduction of cystic cavities was not clearly demonstrated. So we decided not to include the citation in the revised manuscript.

Comment 10) Line 338-339 – the sentence “Thus, it is conceivable that I-5 injection promotes ECM remodeling in the rat spinal cord in a way that emulates the characteristics of the ECM in the mouse spinal cord.” Is both unclear and overly speculative.

Response: According to the reviewer's suggestion, we corrected the sentence to sound more distinct or explicit as shown below (page 19, line 13),

“Thus, it is conceivable that the potential mechanism by which I-5 injection promotes ECM remodeling in the rat spinal cord may be similar to that occurring in the formation of fibrotic scars in the mouse spinal cord.”

Comment 11) It is important for many reasons that the entire study can be reproduced by others. While many of the IHC work is standard and can be performed readily, the most challenging issue with respect to reproduction is the synthesis of the polymer. I am not convinced that all the information provided with this respect has been provided. No value for the polymer concentration was provided. How was it sterilized – filtration? The reaction scheme in Figure 1 could be much clearer, and the HNMR. It appears that while some information is provided, not enough is provided to truly let some other group reproduce this research.

Response: We admit that information on synthesis of I-5 was not provided enough in the original manuscript. In the revised manuscript, we did our best to add detailed contents related to the I-5 synthesis.

The final concentration of I-5 was 10 wt % of polymer solution. The polymer solution was sterilized by filtration using 0.2 μm cellulose acetate syringe filter. This information was added with the separate subheading “Preparation of I-5 hydrogel solution” in the methods section (page 25, line 5).

In addition, we revised figure 1 adding more specific and accurate conditions for synthetic processes. For example, reaction temperature at each stage of synthesis is provided with the duration of each reaction. Furthermore, the relative number of molecules that are attached to the polymer backbone (poly(dichlorophazene)) was given outside of the parentheses. We also provide the full names of the acronyms used in the synthetic scheme (Fig. 1) in the figure legend to help readers who are not expert at chemistry. We also provide information in the figure legend on to which molecular structures in I-5 the specific peaks in HNMR data in Fig. S1 correspond. We hope that the revised Fig. 1 and Fig. S1 would feel easier to be understood.

Comment 12) My final comment is that the authorship team must provide better

description of synthesis and characterization of the polymer. Polyphosphazenes are notoriously difficult to control in synthesis and the information required for other to repeat the synthesis of the polymer is not accurate nor sufficient. As it stands, this work cannot be reproduced, since it lacks sufficient information on even what the material is – the 18771 Da molecular weight (M_w is never presented as accurately as this – 18.7 kDa or approx. 18 kDa is better due to natural errors in the analysis) shows that this was a $n=1$ synthesis, so reproducibility is a major problem here. What were the results of multiple batches? Furthermore, hydrophilic polyphosphazenes hydrolyze quite quickly, and therefore some stability data is required.

This lack of description/standardization of the hydrogel is such a critical issue, that if not fixed, would lead to me recommend that this study is rejected. It is not satisfactory, that a irreproducible polymer with properties that are the primarily reason for the results, is described in this manner. If the research community are unable to accurately synthesize the polymer, then the authors have amazing results that are unrepeatable to anyone else but themselves. I would ask for a more accurate synthesis description that achieves the minimum standards for publication in even low-impact factor journals. Currently the description is not sufficient for publication in ANY journal.

Response: We are thankful for the reviewer’s insightful comments on the reproducibility issue. We absolutely agree that it is not acceptable to report therapeutic effects of a biomaterial that cannot be reproduced elsewhere. Our lab (Center for Biomaterials led by SCS) has more than 10 years’ experience of synthesizing various poly(organophosphazenes)-based hydrogels with more than 20 publications in highly regarded international journals. The synthetic process of the poly(organophosphazenes) has been optimized in a way that compositions of functional moieties in relation to the polymer backbone are consistently reproduced showing an expected range of sol-gel transition properties.

As the reviewer pointed out, synthesis of polymer hydrogels having exactly the same molecular weight would be almost impossible because polymerization of monomers starts randomly in reactions and is likely to result in a range of molecular weight in

general. We did synthesize different batches of the imidazole-conjugated poly(organophosphazenes) and the molecular weight ranged from approximately 14 KD to 18 KD. In the original manuscript, we presented the molecular weight of the I-5 hydrogel from the most recent batch with which the majority of the experiments in this manuscript were performed. Although the molecular weight varied, we found that physical properties of the hydrogels including the temperature-dependent gel-sol transition behavior, which is a critical for being injectable, were remarkably similar. Poly(organophosphazenes)-based hydrogels with functional conjugations other than imidazole also showed an anticipated range of the gel-sol transition properties regardless of differences in the molecular weight. This suggests that the compositions of side groups imparting amphiphilicity of the hydrogel may be more critical than the molecular weight in determining the physical properties of a certain poly(organophosphazenes) gel. Through our extensive works in synthesis of the poly(organophosphazenes)-based hydrogels, we have established highly detailed protocols that allow consistent compositions of each component critical for the temperature-dependent phase transition. In the revised manuscript, we added the detailed protocols for the I-5 synthesis in the methods section (page 23, line 5). We are confident that following the protocols presented in the revised manuscript would lead to generation of reproducible hydrogels with the physical properties similar to those reported in our study.

According to the reviewer's comment, the molecular weight was given as a range of approximate molecular weight value (14KD to 18 KD) rather than a fixed value with a specified number in the revised manuscript (page 6, line 4). In addition, we provided the detailed protocol in our lab for the I-5 synthesis at the beginning of the methods section.

In regard to the stability issue, we performed *in vitro* stability test using fluorescently labeled hydrogels incubated in PBS solution at 37°C. We found that dissolution or hydrolysis of the I-5 hydrogel occurs quite quickly in this *in vitro* setting so that the hydrogel is almost completely dissolved by 7 days, indicating that the hydrogel would be fully biodegradable. Based on the previous study reporting similar degradation kinetics of poly(organophosphazenes)-based hydrogel between *in vitro* and *in vivo*⁵, we

assume that the I-5 hydrogel injected in the spinal cord would not maintain its gel property longer than several days. Degradation of hydrogel *in vivo* would be more complex, however, because degradation behavior is influenced by other factors such as pH and water content in the tissue environment^{5,6}. We could not test the *in vivo* durability of the I-5 hydrogel in the spinal cord because any remaining gels would turn into a sol-phase when animals and tissues are exposed to cold buffer or fixative that is required for the preservation of morphology in histological analysis.

Our study showed that dynamic interaction between I-5 hydrogel and macrophages infiltrating to the lesion site may be critical in its effects on eliminating cystic spaces. This suggests that long-term durability of the gel-like or solid phase may not be a quality that is absolutely required for the bridging function of I-5. In Fig. 4, I-5 injection resulted in enrichment of fibronectin matrix even only 7 days after injection. In addition, we observed a high level of MMP-9 immunoreactivity at 7 days after I-5 injection to an extent similar to that observed at 4 weeks post-injection (data not shown), indicating that the matrix remodeling process has been triggered very early and fully activated only at 7 days post-injection. The fibrotic matrix seems to be solidified replacing potential cystic spaces after that time point, and we cannot contemplate any role of the hydrogel with a gel-like or solid phase during this solidification period. In summary, although we are lacking evidence of sufficient stability of the I-5 hydrogel *in vivo*, we consider that the potential mechanism of tissue repair by I-5 revealed in this study would not require long-term presence of the hydrogel in a gel-like or solid phase.

In the revised manuscript, we inserted the *in vitro* stability test in Fig. 2e.

Comment 13) *Except for the polymer synthesis and rheological characterization using n=1, the number of experiments and their description is statistically satisfactory*

Response: To address the reviewer's concern on no replicate measurement in rheological characterization, we performed 3 more replicate viscosity measurements

(final N = 4). CP-2 viscosity was also measured in four replicates. In the revised manuscript, we presented the range of molecular weight from the 5 batches of I-5 synthesis instead of the fixed molecular weight value from only one batch.

Reviewer #2 (Remarks to the Author):

The paper is a throughout investigation of the effects caused by multiple injections of a synthetic hydrogel into a contusive SCI injury model.

The work is well described however I would suggest some additional experiments to strengthen its impact.

***Comment 1)** Page 5 synthesis section: the authors mention the results of a pilot experiment. It would be helpful for the reader to look at such not published data because comparison with other not satisfactory hydrogels is informative. I suggest to place these data in the SI.*

Response: We are sorry for not being able to follow the reviewer's suggestion. We initially intended to comply with the reviewer's recommendation that the data of the preliminary pilot experiments be placed in the supplementary information. So we reviewed all the available data but found out that most of them were incomplete or lacking detailed information. These preliminary pilot experiments were done more than 6 years ago and all the members in the lab who were involved in this project have left several years before. Therefore, we had no choice but to decide not to include those incomplete data in the manuscript. We screened at least 4 or 5 different hydrogels. Some of them showed inadequate bridging effects or others showed excessive inflammatory reactions.

Comment 2) *Is the hydrogel bioabsorbable? Is it still present at 4 weeks after treatment?*

Response: In previous studies, we have extensively characterized the degradation properties of poly(organophosphazenes)-based hydrogels with functional moieties different from the one used in this study^{5, 7, 8}. Most of the hydrogels showed biodegradability either *in vitro* or *in vivo*, or both.

To address the degradation issue, we performed *in vitro* stability test where fluorescently labeled hydrogel was serially observed during incubation at 37°C and found that I-5 was almost completely degraded by 7 days *in vitro*. Based on the previous study reporting similar degradation kinetics of poly(organophosphazenes) between *in vitro* and *in vivo*⁵, we assume that I-5 hydrogel would not persist by 4 weeks after injection. Degradation kinetics of hydrogel *in vivo* would be more complex, however, because degradation behavior is influenced by other factors such as pH and water content in the tissue environment^{5, 6}. Based on the histological assessment, we could not observe any gel-like materials in the spinal cord tissue with I-5 injection. It would be impossible to demonstrate the presence of I-5 materials because the tissue processing for histologic assessment inevitably requires being exposed to a cold buffer or fixative and I-5 hydrogel, if any remains, would turn into sol-state at cold temperature. To summarize, it is highly likely that I-5 hydrogel becomes biodegraded by 4 weeks after treatment, although we cannot provide definitive experimental evidence for that.

We inserted the *in vitro* stability test data in Fig. 2d.

Comment 3) *Did the author run any previous mass loss test in vitro? If so, a measurement of the bioabsorption time would be interesting. Is the bioabsorption influenced by the chosen functionalization? I think so but authors should better explain that.*

Response: To address the issue of bioabsorption or biodegradation, we performed an additional experiment to observe the *in vitro* stability of I-5 hydrogel at 37°C for both I-5 and CP-2. We found that both hydrogels persist by 4 days but degrade by 7 days *in vitro*.

Our previous studies showed that different functional moieties or side groups attached to the polymer backbone affected the degradation or bioabsorption kinetics^{5, 8}. For example, we reported that introduction of carboxylic group accelerated hydrolysis of the polymer. As for the I-5 hydrogel, conjugation of 1-3 aminopropylimidazole, which is more hydrophobic than carboxylic group, to the carboxyl group would in theory decrease the degradation rate because adding hydrophobic group can decrease water contact-mediated hydrolysis. As we showed in Fig. S6, however, *in vitro* stability of CP-2 was similar to that of I-5. We speculate that the meager influence of imidazole conjugation to carboxylic group on the degradation kinetics may be due to relatively low conjugation rate of I-5 hydrogel compared to carboxylate. As shown in Fig. 1, portion of imidazole group is 0.01 of one polymer unit. Therefore, it is conceivable that potential influence of relatively hydrophobic imidazole on degradation kinetics may be negligible compared to that of hydrophilic carboxylic acid and that the degradation or hydrolysis behavior may be dominated by the carboxylic acid in the I-5 hydrogel.

Comment 4) *Also they may add a few lines about the possible mechanisms of degradation in vivo.*

Response: We previously studied detailed mechanisms of hydrolytic degradation of poly(organophosphazenes) hydrogel with amino acid esters⁶. According to the data in this paper and another one showing introduction of carboxylic group accelerates hydrolysis⁵, an initiation step of hydrolytic degradation of the poly(organophosphazenes) with carboxylic acid esters is hydrolysis of the pendent ester group generating the corresponding free carboxylic acid. This hydrolysis of carboxylic ester is a major mechanism of degradation. The free carboxylic acid then can attack the polymer backbone and mediate chemical reaction leading to the cleavage into small molecules. *In*

vivo, both the initial hydrolysis and the carboxylic ester-mediated backbone cleavage would contribute together to complete biodegradation.

Comment 5) *In order to erase any doubts about collateral sprouting of the spared nervous tracts, instead of regenerating fibers crossing the lesion, neural tracing assessment is recommended. A possible scenario is intracranial injections (over sensorimotor cortex) of anterograde tracers and assessment of axons labeling after 2 weeks at the site of implant.*

Responses: In accordance with the reviewer's suggestion, we performed an additional experiment in which anterograde tracers were injected into the sensorimotor cortex or rostral spinal cord. AAV8-GFP was injected into the sensorimotor cortex to visualize the corticospinal axon. However, GFP positive corticospinal axons exhibited a significant degree of retraction, ending up with stopping at several hundreds micrometer above the rostral boundary of the FN-rich matrix (Fig. 10c in the revised manuscript). We also injected BDA into upper thoracic spinal cord to label various descending axons (either supraspinal or long propriospinal). We were able to observe axons growing across the boundary in the rostral portion of the FN-rich matrix (Fig. 10c', c'' in the revised manuscript).

We incorporated the above results into Figure 10. We added corresponding texts explaining the results (page 16, line 11) and figure legends. In the methods section, we inserted sentences describing the methods of injecting AAV8-GFP and BDA (page 28, line 2)

Comment 6) *The description of the criteria for the selection of the ROI seems to be quite discretionary but it is a critical parameter that may influence results. Please clarify this point.*

Responses: We admit that the description “The locations of ROIs were carefully determined to be comparable in all the sections analyzed” in the original manuscript was too vague and lacking inclusion of anatomical landmarks to make the ROIs positioned consistently across different spinal cord sections. Therefore, we added sentences providing more detailed description about the location of ROIs in relation to internal anatomy of transverse spinal cord sections. The lateral border of a dorsal ROI was placed immediately medial to the dorsal horn so that the dorsal ROI was located on the dorsal column. A lateral ROI was located just above the line crossing the intermediolateral horn. If the intermediolateral horn could not be identified due to the lesion, the lower border of a lateral ROI was placed just above the transverse midline of the spinal cord section. A ventral ROI was located below the ventral horn. If the ventral horn could not be located due to the lesion, the medial border of a ventral horn was placed 500 μm apart from the vertical midline of the spinal cord section.

We also added similar descriptions on placement of ROIs for quantification of MBP immunoreactivity as shown below,

“The ROIs were located using the same criteria as the above ones for quantification of Iba1-immunoreactive signal intensity. The only difference was that they were placed within the MBP-positive residual white matter.”

Comment 7) Figure 2a: what is the kinetic of the gelation process? A viscosity vs time graph at fixed 37 C would be appreciated instead of a qualitative set of pictures.

Responses: We are grateful for the reviewer’s thoughtful suggestion. In accordance with the reviewer’s comment, we performed an addition experiments where changes in viscosity were measured as a function of time elapsed after the temperature is set as 37°C. Within ten seconds after the temperature was fixed at 37°C., hydrogel solution started to form gel-like material with a viscosity of approximately 50 Pa·s. The viscosity rose very

rapidly thereafter and reached almost 80% of the maximum viscosity value attained at 37°C within 2 min (120 secs).

The quantitative graph was added in Fig. 2d in the revised manuscript and the corresponding descriptions were inserted (page 6, line 15).

Comment 8) Also, the movie is not clear as an opaque material seems to appear in the +4C setting as well. In figure 2b the rod-like material cannot be seen. Please provide an enlarged image or mark the material with a dye so as its shape can be better appreciated.

Response: To address the reviewer's criticism, we decide to remove the Movie showing rapid gelation in the revised manuscript. As the reviewer suggested, instead, we provided an enlarged image clearly showing accumulation of gel-like material in a solution at 37°C in Fig. 2b..

Comment 9) Figure 8: please add the BBB score of the sham group. Also the authors should better clarify if the treatment was given at day 0 or at day 7 of the timescale. Lastly, please indicate in the graph the BBB value of the animals after injury and right before treatment.

Response: According to the reviewer's suggestion, the BBB score of the sham group was added to the graph in Fig. 8A. We also added an arrow indicating the day of PBS or I-5 injection.

The BBB data presented on the 7th day were obtained right before injection on that day. This was clarified in the methods section (page 36, line 17). We did not measure BBB scores immediately after injection because we thought that surgical procedures related to the injection and anesthesia could influence animals' performance in overground walking.

Reviewer #3 (Remarks to the Author):

Review Nature Communications: NCOMMS-16-26305

An injectable hydrogel enhances tissue repair after spinal cord injury by promoting extracellular matrix remodeling

Reviewer's summary:

The manuscript describes the use of an injectable hydrogel to reduce cystic cavitation following spinal cord injury. The authors shortly describe the synthesis and main characteristics of the hydrogel prior to investigating its effect on cavity formation in a contusion type injury. They observed a replacement of the fluid filled cavity with fibronectin rich, GFAP-negative scar tissue and an increase in spared tissue. The remodeling of the injury site was partly mediated by MMP9, which was upregulated following hydrogel injection. Blocking the effect of MMP9 with siRNA lead to an increase in cavity size. The authors further demonstrate that the positive effects were mediated by an imidazole moiety, as hydrogels lacking this group demonstrated less positive effects.

Reviewer's comments:

I enjoyed reading the manuscript and the development of injectable hydrogels to promote tissue repair in the central nervous system is an interesting topic and well suited for the readers of nature communications. The authors demonstrated nicely that MMP9 and the imidazole moiety are positively involved in scar formation and it is interesting that the fibrotic scar can lead to neuroprotection and functional improvements.

From my point of view, three main aspects could be further developed to strengthen the manuscript:

Comment 1) The scar tissue formed after injury is an improvement over the fluid filled cavity, but the authors mention that it is mainly composed of fibroblasts and macrophages. Plotting the pathological tissue, white matter and gray matter additionally to the cavity size would be interesting to gain more insight on the tissue response following injury and treatment.

In addition, calculating the volume enclosed by host astrocytes could give further information on the ratio of neural vs scar tissue.

Response: According to the reviewer's suggestion, we calculated 3D volumes of pathologic tissue and residual white matter and added the graphs in Fig. 3g,h. We did not calculate the volume of gray matter though, because the intact gray matter could be delineated only at the regions quite distant from the lesion core.

In addition, we also calculated the volume enclosed by host astrocytes. Because there were remaining tissue sections in only one out of 6 animals with I-5 injection that were included in the original quantification analysis, we generated two more animals each for I-5 and PBS injection groups. Therefore, quantitative volume measurements above included the two more animals newly generated for the revision experiments. Based on the data from these three animals, we found that the volume enclosed by astrocytes (1.02 mm^3) was very similar to cavity volume in PBS injected group. This suggests that cystic spaces supposed to be formed after contusion injury are filled by newly generated matrix enclosed by host astrocytes in animals with I-5 injection. The volume of myelinated white matter was 12.3 mm^3 , so the ratio of neural vs scar tissue was higher than 10-fold, indicating that the amount of neural tissue outnumbered that of scar tissue in animals with I-5 injection.

Comment 2) M2, rather than M1, macrophages have been thought to be beneficial for tissue remodeling. The author could provide further insight on the observed remodeling process by analyzing if there was a shift towards M2 macrophages (e.g. CD206) at the lesion site following hydrogel injection.

Response: According to the reviewer’s suggestion, we performed immunohistochemical staining with anti-CD206 antibodies. Consistent with the reviewer’s suggestion, the majority of macrophages within the fibronectin matrix were positive with CD206. We added the CD206 immunostaining data in Fig. S3 and the relevant texts in the appropriate part of the revised manuscript (page 11, line 1).

We also briefly mentioned implication of CD206-positive macrophages in wound healing mechanisms in the discussion section (page 20, line 8) as shown below,

“The presence of macrophages positive with CD206, a marker of M2 polarization, supports this notion.”

Comment 3) How did the authors conclude that the axons found in the caudal spinal cord were reinnervated rather than spared axons? The differentiation seems to be hard to make.

Responses: To address the reviewer’s question on the issue of reinnervation vs sparing of 5-HT axons, we generated new animals that were sacrificed 7 days after injury or 2 weeks after injury. The animals sacrificed 2 weeks after injury were injected with either PBS or I-5 at 7 days after injury. We reasoned that if the higher axon density at the 8-week time point in I-5 injection group was due to sparing of existing axons, the 5-HT axon density would decrease in PBS group between the 1- and 2-week time points while the density would be maintained or at least decrease to a lesser degree in I-5 group during the same period. However, we found that the 5-HT axon density was sharply decreased at the 1-week time point compared to sham control, and there was no further decrease between the 1- and 2-week time points in PBS group. This suggests that the loss of 5-HT innervation in the lumbar spinal cord was already at a maximum at 1 week after injury. On the other hand, injection of I-5 at the 1-week time point did not significantly influence the 5-HT axon density at the 2-week time point. In PBS group, the extent of 5-HT

innervation at the 1- and 2-week time points was similar to that at 8 weeks, indicating that without I-5 injection, there was no further loss of 5-HT axons or spontaneous reinnervation. Collectively, these results from the additional experiments suggest that 5-HT axons in the lumbar spinal cord are being rapidly lost for the initial 7 days and there would be no further denervation afterwards and that I-5 injected at the 7 day time point would not be involved in sparing of axons destined to be degenerated. Therefore, the higher axon density at 8 weeks after injury in animals with I-5 injection is most likely due to growth of new 5-HT axons reinnervating the ventral motor regions at the lumbar level. We described “reinnervation” rather than “regeneration” because there is no way to distinguish new growth of 5-HT axons from the injured axonal tips from sprouting of 5-HT axons spared in the residual white matter at the epicenter into the gray matter in the lumbar spinal cord.

We incorporated these data from the additional experiments into figure 10d-f. We inserted corresponding texts and figure legends in appropriate places in the revised manuscript (page 17, line 4). In the discussion section, we added several sentences describing the above speculation on the issue of reinnervation vs sparing as shown below (page 21 line12),

“Furthermore, we found an increase in 5-HT axon density in the lumbar spinal cord by I-5 injection, which may have direct relevance with the locomotor recovery. Innervation of 5-HT axons up to 2 weeks after injury was markedly reduced, correlating with the locomotor deficits at this time point. **The substantial increase of 5-HT axons between the 2- and 8-week time points suggests that the FN-matrix formation by I-5 injection supports reinnervation** rather than sparing of existing axons in the ventral motor regions at the lumbar level.”

Comment 4) Furthermore, it would be interesting to demonstrate that axon cross the fibronectin rich area, this could be done using e.g. retrograde tracing.

Responses: Although the reviewer recommended retrograde tracing, we decided not to perform retrograde tracing experiment for the two reasons. First, frequently used retrograde tracers such as FluoroGold and FastBlue do not directly visualize axons that transport the tracers retrogradely, rather they visualize neuronal cell bodies connected to the axons of which tips uptake the tracers. Second, the contusion injury is incomplete, i.e. there is always a significant amount of spared tissue. Retrograde tracers can be transported via axons crossing the FN-rich matrix but also can be transported via axons in the residual white matter spared from the injury. Therefore, presence of neurons positive with retrograde tracers rostral to the FN-rich matrix would not necessarily indicate the transport via crossing axons with the matrix.

Instead of retrograde tracing, we performed anterograde tracing to directly visualize axons growing into the matrix. We injected two tracers in the same animal. First, AAV8-GFP was injected into the sensorimotor cortex to visualize the corticospinal axon. Second, biotinylated dextran amine (BDA) was injected into upper thoracic spinal cord to label various descending axons (either supraspinal or long propriospinal). Although we could not detect corticospinal axons regenerating into the FN-rich matrix, we observed BDA-traced axons within the matrix. The amount of BDA-traced axons decreased at the center of the matrix compared to the boundary region, and we could not detect BDA-traced axons below the FN-rich matrix. So we could not demonstrate axons crossing the FN-rich matrix. Considering importance of neuron-intrinsic factors regulating competence of axon regeneration⁹, however, the failure for descending axon to cross the FN-rich matrix may not be ascribed solely to the property of the matrix in terms of permissiveness to axon growth. If the I-5 hydrogel is used in conjunction with a potential intervention that enhances neuron-intrinsic growth capacity in the future, FN-rich matrix induced by I-5 would permit descending axons to cross the entire length of the matrix.

Additional points include:

Comment 5) The swelling of the material is not mentioned, if the swelling was characterized previously, a simple reference would be sufficient.

Response: We previously characterize the swelling behavior of the poly(organophosphazenes) gel⁸, but not for this I-5 hydrogel. During the *in vitro* stability test inserted in Fig. 2d, we observed slight swelling of I-5 hydrogel at 1 and 2 days *in vitro*. In the revised manuscript, we briefly mentioned about the swelling as shown below (page 6, line 19),

“In a solution set at 37°C, I-5 hydrogel seemed to be swollen at 1 or 2 days *in vitro*, and then started to dissolve by 4 days but still persist at that time.”

Comment 6) How was decision made to inject 10 µl?

Response: In our previous experiments with a different hydrogel¹⁰, we tested different volumes of hydrogel in rat contusion model, although the exact data were not included in the previous paper. We experienced expansion of cavity spaces after injection of the hydrogel in a volume larger than 10 µl. Especially, 20 µl injection resulted in marked enlargement of cystic cavities. Therefore, we decided to start with 10 µl injection first, and it turned out to work just well. So, we decided to keep this volume. It is possible that a volume of less than 10 µl might be sufficient. However, we did not test volumes less than 10 µl because of difficulty in handling smaller volume of hydrogel solution, which is slightly more viscous than water even at 4°C.

Comment 7) Fibroblasts express fibronectin, as demonstrated, which can promote

axonal outgrowth. Additionally, they express many growth factors, which could have acted neuroprotective. This could be discussed.

Response: We agree that fibronectin can be permissive to axonal outgrowth. Fibronectin is a well-known substance for cell adhesion, and numerous studies have shown that neurons extend neurites on an FN substrate¹¹. Moreover, implantation of biomaterials made of FN supports axonal growth in vivo^{12, 13}. Thus, it is likely that FN matrix observed in our study may be supportive of axonal ingrowth. As the reviewer suggested, fibroblasts can be a source of a variety of growth factors. In addition, FN can provide neuroprotective effects via multiple signaling pathways^{14, 15}.

The above discussions were included in the discussion part as shown below (page 21, Line 20),

“On the other hand, FN is a well-known substrate for cell adhesion, and previous studies have shown that FN matrix can support axon growth in in vivo spinal cord injury model. Moreover, FN can provide neuroprotective effects via multiple signaling pathways. Therefore, the formation of fibrotic ECM, which shares a similar mechanism to fibrotic scarring, may contribute to beneficial effects of I-5 hydrogel in tissue preservation and axonal growth.”

Comment 8) *Please plot the irregularity index as a measure of interlimb coordination. (see Koopmans, G.C., Honig, W.M.M., Hamers, F.P.T., Steinbusch, H.W.M., and Joosten, E.A.J. (2005), The assessment of locomotor behavior in spinal cord injured rats; the importance of objective analysis of locomotion. J. Neurotrauma 22, 214-225)*

Response: I'd like to thank the reviewer for the nice information about the parameter of inter-limb coordination. We analyzed the regularity index and added the data. All the necessary parts throughout the manuscript (results, methods, figure legends) were updated to reflect the addition of the regularity index data. In addition, the reference

above was cited in the revised manuscript.

The amount of detail provided for the material and methods seems sufficient for reproduction.

<References cited for the responses to the reviewers' comments>

1. Soderblom C, *et al.* 3D Imaging of Axons in Transparent Spinal Cords from Rodents and Nonhuman Primates. *eNeuro* **2**, (2015).
2. Hill CE, Beattie MS, Bresnahan JC. Degeneration and sprouting of identified descending supraspinal axons after contusive spinal cord injury in the rat. *Exp Neurol* **171**, 153-169 (2001).
3. Ek CJ, *et al.* Spatio-temporal progression of grey and white matter damage following contusion injury in rat spinal cord. *PLoS One* **5**, e12021 (2010).
4. Cho JK, Hong JM, Han T, Yang HK, Song SC. Injectable and biodegradable poly(organophosphazene) hydrogel as a delivery system of docetaxel for cancer treatment. *J Drug Target* **21**, 564-573 (2013).
5. Park M-R, Cho C-S, Song S-C. In vitro and in vivo degradation behaviors of thermosensitive poly(organophosphazene) hydrogels. *Polymer Degradation and Stability* **95**, 935-944 (2010).
6. Lee SB, Song S-C, Jin J-I, Sohn YS. A New Class of Biodegradable Thermosensitive Polymers. 2. Hydrolytic Properties and Salt Effect on the Lower Critical Solution Temperature of Poly(organophosphazenes) with Methoxypoly(ethylene glycol) and Amino Acid Esters as Side Groups. *Macromolecules* **32**, 7820-7827 (1999).
7. Kim YM, Park MR, Song SC. Injectable polyplex hydrogel for localized and long-term delivery of siRNA. *ACS Nano* **6**, 5757-5766 (2012).

8. Seo BB, Koh JT, Song SC. Tuning physical properties and BMP-2 release rates of injectable hydrogel systems for an optimal bone regeneration effect. *Biomaterials* **122**, 91-104 (2017).
9. He Z, Jin Y. Intrinsic Control of Axon Regeneration. *Neuron* **90**, 437-451 (2016).
10. Kang YM, Hwang DH, Kim BG, Go DH, Park KD. Thermosensitive Polymer-based Hydrogel Mixed with the Anti-inflammatory Agent Minocycline Induces Axonal Regeneration in Hemisected Spinal Cord. *Macromolecular Research* **18**, 399-403 (2010).
11. Reichardt LF, Tomaselli KJ. Extracellular matrix molecules and their receptors: functions in neural development. *Annu Rev Neurosci* **14**, 531-570 (1991).
12. King VR, Alovskaya A, Wei DY, Brown RA, Priestley JV. The use of injectable forms of fibrin and fibronectin to support axonal ingrowth after spinal cord injury. *Biomaterials* **31**, 4447-4456 (2010).
13. King VR, Henseler M, Brown RA, Priestley JV. Mats made from fibronectin support oriented growth of axons in the damaged spinal cord of the adult rat. *Exp Neurol* **182**, 383-398 (2003).
14. Sakai T, *et al.* Plasma fibronectin supports neuronal survival and reduces brain injury following transient focal cerebral ischemia but is not essential for skin-wound healing and hemostasis. *Nat Med* **7**, 324-330 (2001).
15. Tate CC, Garcia AJ, LaPlaca MC. Plasma fibronectin is neuroprotective following traumatic brain injury. *Exp Neurol* **207**, 13-22 (2007).

Reviewers' comments:

Reviewer #1 (Remarks to the Author):

The manuscript has become more robust and easier to reproduce as a result of the additional text and experimental work. I appreciate that the authors took the time to consider my initial comments and perform independent experiments and reanalyze the IHC sections. I feel that the manuscript is improved and recommend publication in the current form.

Reviewer #2 (Remarks to the Author):

The manuscript has been significantly improved.
However I have a few more corrections/clarifications for the authors:

from previous comment (2)

The in vitro stability test data was not inserted in Fig. 2d but in Fig. 2e. Please correct.
I agree with the authors that in vitro and in vivo biodegradation times are likely to be different. But from my experience I usually find out that in vivo biodegradation is faster than in PBS solutions because of the presence many additional "degrading factors" like enzymes, macrophages, low-PH and so on. Therefore it may be likely that the I-5 gel will be degraded in a few days. How do the authors justify the observed beneficial effect of their fast degrading matrix over the 2 months follow-up of this study?

from previous comment (5)

Did the author observed any crossing of the BDA stained axons into the caudal boundary of the lesion? Other papers showed promising labeled axons ingrowth but they failed to testify any further migration beyond the site of injury.

from previous comment (7)

please add a few more data points where the viscosity reaches a plateau value (around 120 sec??). It is informative for the reader.

Reviewer #3 (Remarks to the Author):

The authors gave satisfying responses to my questions and included sufficient new data to demonstrate their answers where appropriate.
No further revisions are required from my point of view.

Responses to the reviewers' comments

Reviewer #2

The manuscript has been significantly improved.

However I have a few more corrections/clarifications for the authors:

Comment 1) from previous comment (2)

The in vitro stability test data was not inserted in Fig. 2d but in Fig. 2e. Please correct.

Responses: The above error was made in the response letter. We made sure in this revision that the in vitro stability test data were inserted in Fig. 2e including the figure, figure legend, and text.

I agree with the authors that in vitro and in vivo biodegradation times are likely to be different. But from my experience I usually find out that in in vivo biodegradation is faster than in PBS solutions because of the presence many additional “degrading factors” like enzymes, macrophages, low-PH and so on. Therefore it may be likely that the I-5 gel will be degraded in a few days. How do the authors justify the observed beneficial effect of their fast degrading matrix over the 2 months follow-up of this study?

Responses: We absolutely agree with the reviewer that the in vivo degradation would probably be faster. Accordingly, we do not think that structural scaffolding by I-5 make major contributions to its beneficial effects on abrogation of cystic spaces. Instead, our working hypothesis is that the hydrogel induces or catalyzes complex communications among cellular elements, FN-positive ECM, and matrix remodeling enzymes. Our data suggest that this may be achieved by interaction between I-5 hydrogel and macrophages, a central player of wound healing responses. This interaction probably occurs early after injection while I-5 hydrogel still maintains its gel property and may help prolonged presence of macrophages in the lesion epicenter. The once-triggered macrophage-

mediated wound healing mechanism would persist weeks after while I-5 hydrogel becomes degraded within several days.

To address the above issue in the discussion section, we revised the later part of the 4th paragraph as shown below (from the 13th line in page 20 in the revised manuscript),

“Therefore, macrophages stabilized at the lesion site by I-5 hydrogel may play a central role in the complex communications between MMP-9, perivascular fibroblasts, and FN matrix. Our in vitro stability test suggests that the I-5 hydrogel would be degraded within one week and it is highly likely that in vivo degradation would be faster. We propose that the major function of I-5 hydrogel in our model was to trigger dynamic interaction with macrophages early after injection and thereby activate macrophage-mediated wound healing responses and fibrotic ECM remodeling in the ensuing period.”

Comment 2) from previous comment (5)

Did the author observed any crossing of the BDA stained axons into the caudal boundary of the lesion? Other papers showed promising labeled axons ingrowth but they failed to testify any further migration beyond the site of injury.

Responses: We could not observe evidence of crossing of the BDA-traced axon over the caudal boundary.

The reasons for the lack of axon regeneration beyond the caudal border could be complex. Descending axons traced by BDA, majority of which would be from brainstem motor centers, would have very low intrinsic capacity of regeneration in adult CNS. BDA tracing visualize only a small percentage of axons and tracing efficiency decreases as the distance of injection increases. Moreover, axons that grow into growth-permissive scaffold or matrix would not grow out of the permissive region into the host tissue, which is considered as inhospitable to axon growth. Therefore, we do not think that the lack of crossing axons beyond caudal border would be contradictory to our finding that the

fibrotic matrix was permissive for axon growth.

To address the reviewer's question on this issue, we clearly stated the finding of no crossing of BDA-traced axons over the caudal boundary as shown below (the first line in page 17 in the revised manuscript),

“However, there was no BDA-traced axons regenerating beyond the caudal border of the FN-rich matrix.”

Comment 3) from previous comment (7)

please add a few more data points where the viscosity reaches a plateau value (around 120 sec??). It is informative for the reader.

Responses: According to the reviewer's recommendation, we performed an additional experiment in which the viscosity of the I-5 hydrogel was measured up to 200 seconds after being set at 37°C.

In the revised manuscript, we changed figure 2d with the one reflecting the addition of the new data.

REVIEWERS' COMMENTS:

Reviewer #2 (Remarks to the Author):

The authors satisfactory addressed my concerns. I recommend the publication of the manuscript.